# Learning Urban Climate Dynamics via Physics-Guided Urban Surface-Atmosphere Interactions

Jiyang Xia[*1,2]    Fenghua Ling[2]    Zhenhui Jessie Li[3]    Junjie Yu[1]    Hongliang Zhang[4]
David O. Topping[1]    Lei Bai[2]    Zhonghua Zheng[1]

[1] The University of Manchester [2] Shanghai AI Laboratory
[3] Yunqi Academy of Engineering [4] Fudan University

{jiyang.xia, junjie.yu, david.topping, zhonghua.zheng}@manchester.ac.uk
{lingfenghua, bailei}@pjlab.org.cn   jessielzh@gmail.com   zhanghl@fudan.edu.cn

## Abstract

Urban warming differs markedly from regional background trends, highlighting the unique behavior of urban climates and the challenges they present. Accurately predicting local urban climate necessitates modeling the interactions between urban surfaces and atmospheric forcing. Although off-the-shelf machine learning (ML) algorithms offer considerable accuracy for climate prediction, they often function as black boxes, learning data mappings rather than capturing physical evolution. As a result, they struggle to capture key land-atmosphere interactions and may produce physically inconsistent predictions. To address these limitations, we propose UCformer, a novel multi-task, physics-guided Transformer architecture designed to emulate nonlinear urban climate processes. UCformer jointly estimates 2-m air temperature ($T$), specific humidity ($q$), and dew point temperature ($t$) in urban areas, while embedding domain and physical priors into its learning structure. Experimental results demonstrate that incorporating domain and physical knowledge leads to significant improvements in emulation accuracy and generalizability under future urban climate scenarios. Further analysis reveals that learning shared correlations across cities enables the model to capture transferable urban surface–atmosphere interaction patterns, resulting in improved accuracy in urban climate emulation. Finally, UCformer shows strong potential to fit real-world data: when fine-tuned with limited observational data, it achieves competitive performance in estimating urban heat fluxes compared to a physics-based model. [1]

## 1 Introduction

Occupying only ~3% of the Earth's land surface [27, 31], urban areas concentrate more than 50% of the global population [30], emit ~70% of total greenhouse gases [1], and represent the primary settings where humans experience the impacts of climate change [46]. This disproportion between the climatic importance of cities and their limited representation in global land cover highlights the pressing need to understand and predict urban climate. Nevertheless, urban climates remain largely underrepresented in machine learning (ML)-based climate models [7, 12, 23]. A primary reason is that urban areas constitute only ~8% of grid cells in coarse-resolution global climate models (0.9° × 1.25°), leading to their marginalization in most ML-based approaches. Moreover, urban climate modeling poses unique challenges that are not captured by these approaches.

---

[*]This work was done during his internship at Shanghai Artificial Intelligence Laboratory.
[1]The code and datasets of this work are available at https://github.com/envdes/code_UCformer.

39th Conference on Neural Information Processing Systems (NeurIPS 2025).

Urban climate modeling is particularly challenging due to the heterogeneous nature of urban surfaces. Urban environments feature complex structures such as street canyon morphology, varied surface thermal and radiative properties, and irregular land use configurations [11]. Such diversity significantly affects local energy exchanges and atmospheric coupling [25]. These complexities remain difficult to capture with existing ML-based climate models. Despite advances in spatial resolution, most models rely on purely data-driven mappings and lack representation of underlying physical processes [41, 44]. These limitations bring two key learning challenges for ML-based urban climate models: (1) learning is often inefficient due to the complexity and variability of urban environment, and (2) limited physical interpretability hinders their generalization to different urban climate scenarios.

A physics-guided modeling approach, particularly one that learns physics-based models, appears to be a compelling solution to these challenges. The physics-based models enable the physically interpretable representation of urban climate processes [19, 32, 5, 38], which can guide the learning process in ML approaches. Encoding the interaction between urban surface-atmosphere is crucial to capture the challenge posed by the complexity of urban environment (Fig. 1). Moreover, these interactions drive the evolution of surface energy fluxes, which are further modulated within the urban canyon and collectively underpin the surface energy balance, providing a physical foundation for predicting urban climate variables.

Here, we propose UCformer, a physics-guided deep learning architecture tailored for local urban climate modeling. UCformer incorporates physical and climatic knowledge through two carefully designed components: (1) a domain-specific encoder that dynamically encodes urban surface-atmosphere processes, capturing the modulation of atmospheric forcing by surface features and embedding surface fluxes as latent variables; (2) a physics-guided decoder that embeds inductive biases derived from physical laws, allowing the learning process to capture variable interdependence as an emergent property rather than as isolated prediction targets. Extensive experiments show that UCformer achieves superior performance in multi-task urban climate estimation, with a 12.7% relative improvement in urban temperature prediction accuracy and a 13.4% gain in

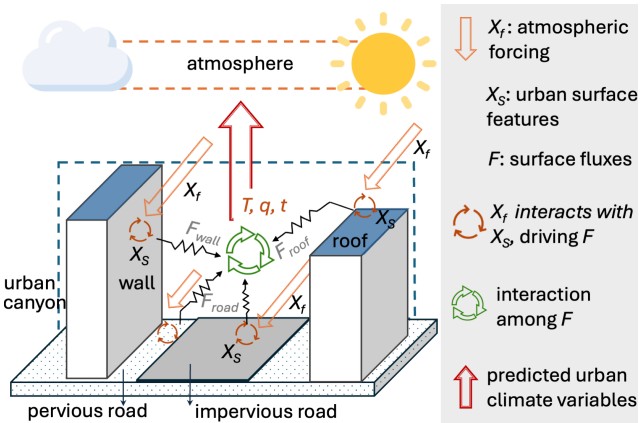

Figure 1: Urban surface–atmosphere interactions in urban climate system. Atmospheric forcing is modulated by urban surface characteristics, and urban climate states are diagnosed through the surface energy balance. We encode $X_f$ and $X_s$ as inputs, model fluxes $F$ as latent representations, and estimate $T$, $q$, and $t$ as outputs.

generalization to future urban climate dynamics compared to the best baseline. Furthermore, learning a neural model based on simulated data cuts city-scale climate simulation (55 years) time from ~12 hours to ~17 seconds, making large-scale scenario analysis practical and efficient (discussed in Appendix A.1). Additionally, despite being trained on physics-based simulations, UCformer shows strong potential to fit real-world data. While our solution is motivated by urban climate modeling, the underlying design principles are grounded in general surface-atmosphere interaction mechanisms, which can be shared across other Earth science domains.

We summarize our contributions and findings as follows:

- We introduce a comprehensive ML-ready dataset tailored for urban climate modeling, filling a critical gap in existing climate benchmarks. The dataset captures detailed urban surface characteristics, atmospheric forcing, and physics-based outputs, and is designed to facilitate ML research on the unique challenges posed by urban climate systems.

- We propose a physics-guided modeling approach that integrates domain and physical knowledge into the neural network to represent urban surface-atmospheric processes, enabling excellent accuracy of local urban climate estimations.

- With its domain-specific encoder and physics-guided decoder, our model effectively generalizes future urban climate dynamics, excels at multi-task estimations, and extends well to sparse real-world data. This physics-guided design presents a transferable concept for broader Earth system sciences, demonstrating strong potential for tackling complex multi-task learning and modeling sparse data.

## 2  Related work

**ML for urban climate modeling.** Urban climate modeling simulates interactions between atmospheric forcing and urban surfaces characterized by long-term averages and distributions. Multi-model ensemble climate projections that incorporate urban surface energy processes are crucial for climate-informed urban development [22]. However, such projections are universally unavailable and challenging to implement [22]. Zhao et al. [46] first fills this gap by developing a reduced-order regression model to generate local urban climate projections under different scenarios, facilitating the comprehension of the impacts of climate change on local urban climates. Zheng et al. [47] expands the work of Zhao et al. [46] by introducing non-linear regression models to map the local response of urban climate to atmospheric forcing. Meyer et al. [22] develops an urban neural network to estimate radiation and fluxes reflected to the atmosphere following the effects of urban parameterization, enabling it to be coupled to numerical models and thereby reducing computational demands. However, these models primarily focus on learning data mappings, while often overlooking the underlying physical evolution, such as the influence of urban surface characteristics on surface energy processes (discussed in Appendix A.2), as well as the physical relationships between urban climate variables. Moreover, the generalizability of these methods remains insufficiently explored, particularly in the context of urban climate dynamics over long time spans.

**Physics-guided ML.** Concerns regarding the generalizability and interpretability of ML have led to the development of new machine learning strategies in some scientific domains, known as physics-guided ML or physics-informed ML [28, 41, 39]. For instance, Read et al. [29] introduced a physics-guided loss (a simplified energy budget formulation) that penalizes ML predictions violating energy conservation, enabling their long short-term memory model to better generalize to unseen scenarios. Some studies, however, have shown that physical loss function does not consistently improve model generalization [44, 6]. In contrast, physics-guided architecture embeds physical meaning into neural network neurons or incorporates domain-specific knowledge into model structures to achieve task-specific designs [9, 3]. Drawing on the relationship between predictands, Zanetta et al. [44] changed the neural network architecture by adding an equation layer to derive two additional predictands which ensure the predictands obey the physical laws and thus enhance the model interpretability. Such strategies enhance model interpretability and generalizability, and hold the potential to advance modeling of the interactions between urban environments and atmospheric processes. Building on the inspiration provided by these methods, our work explores a physics-guided ML model tailored to urban climates, refining the understanding of urban climate projection.

## 3  Methods

### 3.1  Problem formulation

The urban climate modeling problem can be summarized as a mapping from a set of atmospheric forcing $X_f$ to a set of urban climate predictands $\hat{I}$ given the local urban surfaces characteristics ($X_s$):

$$F_\theta(\hat{I}) = P(X_f|X_s), \tag{1}$$

where $\theta$ represents the parameters of the model and the atmospheric forcing $X_f \in \mathbb{R}^{T \times C \times N}$ consists $C$ variates modeled at $N$ different cities. $T$ denotes an interdependency time sequence, where the simulation of urban climate is influenced by previous timesteps. Currently, this interdependency is omitted at the daily scale [20, 46]. To fill the gap in fine-grained temporal urban climate projections, this work extends the timestep to every 3 hour, making it essential to capture dependencies between timesteps. For more details, see Appendix A.3.

## 3.2 UCformer overview

Drawing inspiration from the urban surface-atmosphere interaction in some physics-based models [21, 25], we strategically design this variant Transformer for urban climate emulation, named UCformer. As shown in Fig. 2, instead of stacking and embedding all features together, we embed atmospheric forcings $X_f$ and urban surface parameters $X_s$ into two independent embedding blocks (each forming an independent sequence), respectively. This embedding strategy accounts for the inherent differences between the two types of data, while enabling seamless integration with the encoder component. The encoder of our model functions as urban climate process adaptive operators described in Sec. 3.3, which consists of two blocks. The first encoder block learns the modulation of atmospheric forcing by urban surfaces, reflecting their role in shaping urban climate dynamics. The subsequent blocks focus on learning and simulating the flux iterative calculation processes within the urban canyon. Urban 2-m air temperature ($T$), specific humidity ($q$), and dew point temperature ($t$) are subsequently diagnosed via the physics-guided decoder, which embeds inductive biases derived from governing physical laws that characterize the intrinsic relationships among the target variables.

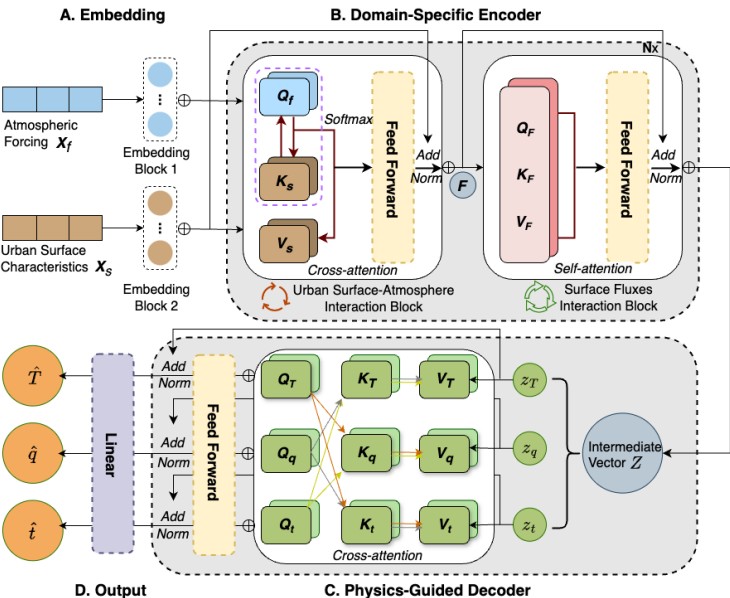

Figure 2: UCformer architecture. A: feature embedding strategy. Atmospheric forcing and urban surface features are separately embedded as query, key, and value matrices. B: The structure of the domain-specific encoder, which contains the urban surface-atmosphere interaction block and flux iteration block. C: The structure of the physics-guided decoder. This module enables the model to diagnose target variables by aggregating information from its physically coupled counterparts. D: Predictands are processed by three separate fully connected layers. A component-level mapping bridging the physics-based model and machine learning modules is provided in Appendix A.4.

### 3.3 Domain-specific encoder with urban surface-atmosphere interaction operators

The atmospheric forcing $X_f$ is modulated by the urban surface characteristics $X_s$ at the sub-grid scale in physics-based models. Specifically, the interaction between $X_f$ and $X_s$ can be broadly categorized as follows: firstly, the individual urban surfaces absorb solar and longwave radiation, and there is radiatively interaction between these surfaces. Fluxes from each urban surface interact with each other in the urban canyon (Fig. 1). Then, the temperature ($T$), specific humidity ($q$), dew point temperature ($t$), and other variables output to the atmosphere are predicted. Consequently, we can characterize the above process by the following simplified equation:

$$\hat{I} = f(H, E, G), \tag{2}$$

$$\text{sum}(H, E, G) = \vec{S} - \vec{L}, \tag{3}$$

$$\vec{S} = \sum_{\Lambda} \left[ W_{\text{roof}} \vec{S}_{\text{roof},\Lambda} + (1 - W_{\text{roof}}) \vec{S}_{\text{uc},\Lambda} \right], \text{where } \vec{S}_{\text{roof,uc},\Lambda} = \phi(X_f, X_s), \tag{4}$$

$$\vec{L} = W_{\text{roof}}(L_{\text{roof}}\uparrow - L_f\downarrow) + (1 - W_{\text{roof}})(L_{\text{uc}}\uparrow - L_f\downarrow), \text{where } L_{\text{roof,uc}}\uparrow = \phi(X_f, X_s). \quad (5)$$

Eq. 2 defines the prediction of a set of urban climate variables ($\hat{I}$), based on the sensible flux ($H$), latent flux ($E$), and ground flux ($G$) in the urban canyon [25]. Eq. 3 denotes that the sum of fluxes must be balanced by the net solar radiation ($\vec{S}$) and net longwave radiation ($\vec{L}$) absorbed by the urban canyon [25]. Eq. 4 and Eq. 5 describe the numerical solutions of radiation in the urban canyon [25], where the net solar radiation including visible ($\Lambda$) direct and diffusion radiation of roof ($\vec{S}_{\text{roof},\Lambda}$) and other surfaces ($\vec{S}_{\text{uc},\Lambda}$).

Features $X_f$ and $X_s$ are important to derive the radiation and fluxes required in the urban canyon energy budget balance. We model the processes (Eq. 4 and Eq. 5) through a cross-attention operator to represent the interaction between $X_f$ including solar radiation ($S_f\downarrow$) and longwave radiation ($L_f\downarrow$), and $X_s$, which encompasses properties such as emissivity and albedo, formulated as:

$$\hat{F} = softmax(\frac{Q_f K_s^T}{\sqrt{d_k}})V_s, \quad (6)$$

where $Q_f$ is derived from $X_f$, while the key ($K_s$) and value ($V_s$) matrices are obtained from $X_s$. This assignment reflects the physical principle that atmospheric forcing, such as radiation and wind fields, dynamically responds to heterogeneous urban surfaces, which modulate energy and fluxes. Then the output ($\hat{F}$) of the cross-attention operator goes through serval self-attention operators representing the iterative fluxes interactions (Eq. 2 and Eq. 3):

$$\hat{z} = softmax(\frac{Q_F K_F^T}{\sqrt{d_k}})V_F, \quad (7)$$

where $\hat{z} \in \mathbb{R}^{T \times D \times N}$ is regarded as the latent representative of urban climate predictands with dimensionality $D$.

## 3.4 Physics-guided decoder incorporating soft physical constraints

The physics-guided decoder enables the model to infer each target variable not in isolation, but by leveraging the latent representations of its physically coupled counterparts, imposing a soft form of physical constraint. The variables of interest in this work ($T$, $q$, $t$) are physically coupled through a set of physical equations, including the empirical Magnus–Tetens formula:

$$t = \frac{b \cdot \gamma(T, RH)}{a - \gamma(T, RH)}, \text{where } \gamma(T, RH) = \frac{a \cdot T}{b + T} + \ln\left(\frac{RH}{100}\right), \quad (8)$$

$$RH = \frac{q}{q_s} \cdot 100 = \frac{q \cdot (P - 0.378e_s)}{0.622e_s} \cdot 100, \text{where } e_s = c \cdot \exp\left(\frac{a \cdot T}{b + T}\right), \quad (9)$$

where $a$, $b$ and $c$ are empirical coefficients, and $RH$ represents the relative humidity. Eq. 9 is derived from the ideal gas law for dry air and water vapor. The saturation water vapor pressure ($e_s$) in Eq. 9 follows the identical structure as the August-Roche-Magnus equation [44]. Thus, the physical constraint among $T$, $q$ and $t$ can be described by an implicit equation $F(T, q, t, P) = 0$. However, the absence of pressure ($P$) prevents explicit incorporation of the physical equation into the model architecture or loss function. We propose a more flexible, representation-level approach to encode physical relationships. Rather than treating each variable as an independent prediction target, such as modeling $\hat{t} = Wz_t + b$. Our approach encourages the model to learn physical dependencies as emergent properties. Specifically, we reformulate the prediction of $t$ as:

$$\hat{t} = Wg(z_t, z_q, z_T) + b, \quad (10)$$

where $g(z_t, z_q, z_T)$ denotes a latent functional mapping that captures the interaction among the variables through a cross-attention mechanism. This enables the model to adaptively aggregate information from the latent representations of $T$ and $q$ when predicting $t$, formulated as:

$$g(z_t, z_q, z_T) = \sum_{i=1}^{n} \alpha(Q_{z_t}, K_{\text{concat}(z_q, z_T)})V_{\text{concat}(z_q, z_T)}. \quad (11)$$

According to Eq. 10 and Eq. 11, the physics-guided decoder can be described as:

$$\hat{y_i} = WA_iV_{jk} + b, \quad (12)$$

where $A_i$ is the cross-attention score of variates $j$ and $k$ with respect to target variate $i$.

# 4 Experiments

To evaluate our models, we conduct experiments on both simulation and observational datasets and try to answer the following questions:

- How does the model perform on the local urban climate emulation task and generalize beyond training data, particularly in emulating future climate dynamics and transferring the model to previous unseen cities?
- As a physics-guided ML model, what are the respective roles of the domain-specific encoder and the physics-guided decoder in capturing urban climate processes?
- Does learning shared representations of urban surface–atmosphere interactions across cities lead to better performance than city-specific or locally focused models?
- What is the potential of the physics-guided ML model, developed using simulation data, to extend to real-world settings?

## 4.1 Dataset and setup

**Simulation data.** The simulation datasets are derived from the physics-based Community Land Model Urban (CLMU) [25], offering city-scale urban climate simulations for six cities (see Appendix B.1). These datasets explicitly incorporate urban surface-atmosphere processes and comprise comprehensive atmospheric forcings and detailed urban surface characteristics. The training set consists of data spanning from 2020 to 2044, with 8 time steps per day, totaling 1,095,000 data points. The validation and test sets comprise data from 2045–2049 and 2050–2055, each containing 219,000 data points. We further use the data from 2070 to 2074 to assess the generalizability of our model (The discrepancy in data distribution is detailed in Appendix A.5). Detailed descriptions of the dataset implementation and the variables used for model development are provided in Appendix B.1 and B.2.

**Observational data.** The observational data are sourced from Lipson et al. [19], including observed atmospheric forcing and features surrounding the site. In this work, atmospheric forcing and heat flux data from the "UK-King's College" site in London were used to fine-tune UCformer, investigating its potential to transition from physics-based simulation estimates to real-world data emulations. Specifically, the dataset of UK-King's College contains observational data from 2012-04-04 00:00:00 to 2013-12-31 23:30:00 with a time step of 30 minutes. After the clean process (drop samples with "NaN" value for target labels), this dataset processes 19,725 data points.

**Training objectives.** Deterministic urban climate emulation models can be trained by minimizing the MSE for rolled-out estimates. The model in this work is trained to minimize the aggregated MSE loss for multi-task estimates: $\mathcal{L} = \mathcal{L}_T + \mathcal{L}_q + \mathcal{L}_t$. The objective is to optimize the model parameters $\theta$ by minimizing the total loss $\theta^* = \arg\min_\theta \mathcal{L}$. The model is optimized using the Adam optimizer: $\theta \leftarrow \theta - \eta\nabla_\theta\mathcal{L}$, and the learning rate ($\eta$) is adjusted dynamically using a scheduler: $\eta \leftarrow 0.9\eta$.

**Implementation details.** We implemented the model using PyTorch and finalized its configuration via hyperparameter tuning with Optuna [2] in 35 GPU hours (NVIDIA 4090). The model was trained using the Adam optimizer and Gaussian Error Linear Unit (GELU) activation function, with a batch size of 64, a learning rate set to 1e-5, a training epoch of 50, and a dropout rate of 0.1. Appendix B.3 lists the detailed configuration of models.

## 4.2 Baselines

Since there are no widely adopted models specifically designed for urban climate emulation, we select baselines based on prior work related to urban climate modeling [47, 46] and urban representation [22]. Although the task investigated in this work does not fall strictly under the category of time series forecasting problem where future values are predicted from labeled historical sequences [36, 17], our task involves input variables with inherent temporal structures, particularly atmospheric forcing. In this context, time-series feature representation and modeling become a shared concern between our task and standard forecasting problems [14, 48, 40]. Consequently, representative time series forecasting models are included in the baseline comparisons to enable a more comprehensive assessment of the model's ability to leverage temporal information embedded in the input. Furthermore, physics-guided ML models are excluded from the baseline set due to the absence of methods tailored

to urban climate and the highly data- and task-specific nature of existing approaches, which typically rely on embedding explicit physical or partial differential equations [29, 44] that are not readily applicable in our setting.

Overall, we compare UCformer against several representative baselines: a Multi-layer Perceptron following the top-performing architecture in the ClimSim benchmark [43] (MLP_CSB), an automated machine learning method (AutoML), a vanilla Transformer, and two time-series forecasting models adapted to our framework—Informer_modified [48] and LSTNet_modified [14]. Notably, AutoML estimators are constructed in a task-specific manner, requiring separate training for each prediction target.

## 4.3 Metrics

Two metrics are screened to evaluate the overall performance of models: the root mean squared error (RMSE) and the Mean Emulation Skill Score (MESS) [44]. Here we defined MESS as:

$$\text{MESS} = 1 - \frac{\text{MAE}_{\text{em}}}{\text{MAE}_{\text{ref}}}, \tag{13}$$

where $\text{MAE}_{\text{em}}$ represents the mean absolute error (MAE) of urban climate emulators, while $\text{MAE}_{\text{ref}}$ refers to the MAE between the simulated urban climate and its corresponding atmospheric forcing. The MESS presented here aims to denote the emulation skill of different models as $\text{MAE}_{\text{ref}}$ can be regarded as a lower-bound expectation of an emulator. We might expect that our models will have a value of MESS between 0 and 1 (higher values are better). It also allows an emulator to score a much lower value than 0 if its emulation is much worse than simply using its atmospheric counterpart. For multi-task evaluation, we report the aggregated MESS ($\text{MESS}_{\text{agg}}$) across urban air temperature ($T$), specific humidity ($q$) and dew point temperature ($t$). Statistical significance is assessed via the paired Wilcoxon signed-rank test, as detailed in Appendix A.6.

## 4.4 Main results

**Skillful urban climate multi-task emulations by UCformer.** We report the performance of models on the dataset of 2050-2054 in Tab. 1. UCformer consistently outperforms other models in terms of RMSE and MESS for both single-variable and integrated multi-task emulations. In addition, MLP_CSB ranks second overall, achieving a multi-task emulation score (MESS) of 1.9088, and outperforming both modified sequence forecasting models, LSTNet_modified and Informer_modified. This result indicates that simply modifying the data interface of time-series forecasting models to fit our task may not resolve the underlying mismatch in problem formulation. Such adaptation may impose assumptions about temporal dependencies that are not aligned with the objectives of urban climate emulation, and risk mischaracterizing the nature of the task. The performance gap may also be exacerbated by the extent to which these models are tailored for sequence forecasting. For example, although Informer builds on a Transformer-based architecture with higher model complexity and parameter count than LSTNet (which combines CNN and RNN modules), the lighter LSTNet_modified achieves better results in our task.

Table 1: Performance metrics for multivariate estimation. Bold values indicate the best performance.

| 2050–2054 | | $T$ | | $q$ | | $t$ | |
| method | $\text{MESS}_{\text{agg}}\uparrow$ | RMSE(K)$\downarrow$ | MESS$\uparrow$ | RMSE(g/kg)$\downarrow$ | MESS$\uparrow$ | RMSE(K)$\downarrow$ | MESS$\uparrow$ |
|---|---|---|---|---|---|---|---|
| MLP_CSB | 1.9088 | 0.3846 | 0.8128 | 0.2078 | 0.5524 | 0.1945 | 0.5436 |
| AutoML | 1.8584 | 0.3928 | 0.8088 | 0.2183 | 0.5216 | 0.1993 | 0.5280 |
| Transformer | 1.7173 | 0.3737 | 0.8136 | 0.2444 | 0.4631 | 0.2454 | 0.4406 |
| LSTNet_modified | 1.9065 | 0.3528 | 0.8286 | 0.2238 | 0.5075 | 0.1910 | 0.5704 |
| Informer_modified | 1.7700 | 0.3567 | 0.8268 | 0.2325 | 0.4775 | 0.2003 | 0.4657 |
| UCformer | **2.0468** | **0.3264** | **0.8405** | **0.1851** | **0.6157** | **0.1777** | **0.5906** |

**Generalizing to future urban climate dynamics.** The generalizability of ML models is a widely concerned issue, and physics-guided models have been demonstrated to have physically reliable prediction results while being expected to have better generalization [39]. We explored the temporal generalizability of our model using the dataset of 2070-2074 (out-of-sample) and report the performance in Tab. 2. The evaluation results on the testing dataset (Tab. 1) and the generalization dataset (Tab. 2) highlight the strong temporal generalization ability of the UCformer. UCformer

consistently achieves the lowest RMSE and highest MESS across all three variables in both datasets, outperforming all other models. Notably, in the generalization dataset (2070-2074), the RMSE values of UCformer for all three variables are lower than most of the other models on the testing dataset (2050-2054) for the corresponding variables. Moreover, the probability density function (PDF) comparison in 2070-2074 is presented in Appendix A.7.

Overall, UCformer consistently demonstrates a strong skill to approximate all predictands distributions across a time scale. In contrast, other baselines deliver notable deviations. On this basis, our results highlight the exceptional ability of UCformer to generalize across time scales, showcasing its robustness and adaptability to future scenarios.

Table 2: Temporal generalization performance for multi-variate estimations for future scenarios.

| 2070–2074 | | $T$ | | $q$ | | $t$ | |
|---|---|---|---|---|---|---|---|
| method | MESS$_{agg}$↑ | RMSE(K)↓ | MESS↑ | RMSE(g/kg)↓ | MESS↑ | RMSE(K)↓ | MESS↑ |
| MLP_CSB | 1.8809 | 0.4070 | 0.8031 | 0.2251 | 0.5394 | 0.1994 | 0.5384 |
| AutoML | 1.7995 | 0.4275 | 0.7941 | 0.2517 | 0.4947 | 0.2118 | 0.5108 |
| Transformer | 1.6746 | 0.3910 | 0.8032 | 0.2765 | 0.4348 | 0.2414 | 0.4366 |
| LSTNet_modified | 1.8944 | 0.3703 | 0.8211 | 0.2356 | 0.5450 | 0.2023 | 0.5283 |
| Informer_modified | 1.6585 | 0.3877 | 0.7803 | 0.2194 | 0.4202 | 0.2007 | 0.4586 |
| UCformer | **1.9451** | **0.3385** | **0.8358** | **0.2121** | **0.5671** | **0.1960** | **0.5422** |

**Transferability to unseen cities.** Spatial generalization remains a central concern in the climate science community, given its importance for real-world applicability but also its inherent difficulty for most machine learning models. In this context, we conduct additional "leave-one-city-out" experiments to evaluate the spatial generalization capabilities of UCformer, with results summarized in Tab. 3.

The results indicate that UCformer demonstrates a notable degree of spatial generalization, although its performance exhibits variation across cities, potentially driven by differences in urban morphology and local climate conditions. For example, the temperature ($T$) estimation error increases from approximately 0.02 K in Singapore to 0.07 K in Rome, while the RMSE for dew point temperature ($t$) rises from 0.01 K in Rome to 0.03 K in London. These findings suggest that model spatial generalization may be sensitive to specific urban contexts or climatic conditions, highlighting the need for further investigation into the underlying factors affecting spatial generalization.

Table 3: Performance metrics of UCformer in "leave-one-city-out" experiments (values in parentheses denote results under the original setting).

| 2050–2054 | | $T$ | | $q$ | | $t$ | |
|---|---|---|---|---|---|---|---|
| Held-out city | MESS$_{agg}$↑ | RMSE(K)↓ | MESS↑ | RMSE(g/kg)↓ | MESS↑ | RMSE(K)↓ | MESS↑ |
| London | 1.6629 | 0.2990 | 0.8127 | 0.1344 | 0.4357 | 0.1977 | 0.4144 |
| | (1.9007) | (0.2419) | (0.8541) | (0.1117) | (0.5171) | (0.1655) | (0.5295) |
| New York City | 1.6989 | 0.4167 | 0.7699 | 0.1904 | 0.5063 | 0.2507 | 0.4225 |
| | (1.8901) | (0.3515) | (0.8030) | (0.1645) | (0.5741) | (0.2212) | (0.5130) |
| Shanghai | 1.9369 | 0.4806 | 0.8349 | 0.2886 | 0.5555 | 0.2529 | 0.5465 |
| | (2.0024) | (0.4437) | (0.8486) | (0.2611) | (0.5986) | (0.2426) | (0.5552) |
| Singapore | 2.2840 | 0.2668 | 0.8581 | 0.2798 | 0.7060 | 0.1279 | 0.7198 |
| | (2.4586) | (0.2412) | (0.8727) | (0.2130) | (0.7975) | (0.1036) | (0.7884) |
| Sao Paulo | 1.8953 | 0.3965 | 0.8179 | 0.2587 | 0.5326 | 0.1649 | 0.5448 |
| | (2.1241) | (0.3561) | (0.8363) | (0.2074) | (0.6618) | (0.1435) | (0.6260) |
| Rome | 1.8163 | 0.3972 | 0.7843 | 0.1571 | 0.5197 | 0.1975 | 0.5123 |
| | (1.9048) | (0.3242) | (0.8282) | (0.1477) | (0.5450) | (0.1896) | (0.5316) |

## 4.5 Ablation studies

**Domain and physics-guided neural operators are crucial for urban climate estimations.** To evaluate the contribution of the two proposed components in urban climate emulation, we conducted ablation by replacing the domain-specific encoder and the physics-guided decoder with a vanilla encoder and decoder, respectively. Tab. 4 highlights the impact of these neural operators and illustrates how UCformer outperforms the vanilla Transformer in multi-task learning. Instead of merely learning a direct mapping from $X_f$ and $X_s$ to target variables, the domain-specific encoder (DE) models the evolution of urban climate physical processes. This approach enhances the representation of

urban surface–atmosphere interaction and facilitates local urban climate emulations. Additionally, the physics-guided decoder (PD) further boosts emulation accuracy.

Table 4: Ablation study regarding the effectiveness of the domain-specific encoder (DE) and physic-guided decoder (PD).

| 2050-2054 | Module | | | $T$ | | $q$ | | $t$ | |
|---|---|---|---|---|---|---|---|---|---|
| method | DE | PD | MESS$_{agg}\uparrow$ | RMSE (K)$\downarrow$ | MESS$\uparrow$ | RMSE (g/kg)$\downarrow$ | MESS$\uparrow$ | RMSE (K)$\downarrow$ | MESS$\uparrow$ |
| Transformer | ✗ | ✗ | 1.7173 | 0.3737 | 0.8136 | 0.2444 | 0.4631 | 0.2454 | 0.4406 |
| | ✓ | ✗ | 1.9673 | 0.3389 | 0.8366 | 0.2022 | 0.5713 | 0.1879 | 0.5594 |
| UCformer | ✗ | ✓ | 1.9013 | **0.3176** | **0.8417** | 0.2181 | 0.5441 | 0.1964 | 0.5156 |
| | ✓ | ✓ | **2.0468** | 0.3264 | 0.8405 | **0.1851** | **0.6157** | **0.1777** | **0.5906** |

**Learning shared representations of urban climate processes improves emulation performance.** Urban climates exhibit distinct patterns tied to their geographic context, but may share learnable regularities across cities. UCformer, our proposed model, is by default trained using a multi-city strategy to capture shared surface–atmosphere interaction patterns. To assess the contribution of this shared correlation learning, we introduced a single-city training setting (denoted UCformer_single), where a separate model is trained independently for each city using only its local data. This ablation isolates the effect of inter-city representation sharing on emulation performance.

As shown in Tab. 5, training across multiple cities significantly improves overall urban climate modeling performance. Per-city results are provided in the Appendix A.8, where five cities show notable gains. However, performance declines in Singapore suggest that city-specific training may be preferable for cities with highly uniform climatic conditions. In particular, Singapore exhibits significantly smaller temperature fluctuations throughout the year compared to the other cities, which may limit the benefits of shared representation learning. These results indicate that while shared modeling is generally effective, its utility may vary depending on the climatic distinctiveness of cities.

Table 5: Ablation study on the impact of shared correlation learning in urban climate emulation. The results are first computed on each city individually and then averaged.

| 2050-2054 | Number of Cities | | $T$ | | $q$ | | $t$ | |
|---|---|---|---|---|---|---|---|---|
| method | $N$ | MESS$_{agg}\uparrow$ | RMSE (K)$\downarrow$ | MESS$\uparrow$ | RMSE (g/kg)$\downarrow$ | MESS$\uparrow$ | RMSE (K)$\downarrow$ | MESS$\uparrow$ |
| UCformer_single | $N=1$ | 1.6126 | 0.3941 | 0.8020 | 0.2294 | 0.4368 | 0.2391 | 0.3738 |
| UCformer | $N=6$ | **2.0468** | **0.3264** | **0.8405** | **0.1851** | **0.6157** | **0.1777** | **0.5906** |

## 4.6 The potential from simulation to real world

To address the challenge posed by the scarcity of climatic observations in urban areas, we explored the potential of UCformer to fit real-world data. Since the flux tower observations do not include temperature, humidity, or dew point data, we fine-tuned UCformer on partial real-world data to enable prediction of new predictands (sensible heat flux $Q_h$ and latent heat flux $Q_{le}$) (detailed in Appendix B.4). In Tab. 6, we investigated the importance of data scarcity on UCformer fine-tuning by providing proportional data in chronological order (with the remaining data serving as the testing set). Note that we used the metrics MAE and correlation coefficient ($r$) for evaluation in this section to align with the reference [19]. Overall, the performance of UCformer in emulating $Q_{le}$ and $Q_h$ improves as the amount of fine-tuning data increases.

We conducted a comparative experiment to evaluate how well UCformer can fit real-world data under limited observational conditions, comparing it against the physics-based model CLMU and a baseline trained solely on observations. First, UCformer was fine-tuned using 20% of the available observational data to estimate $Q_{le}$ and $Q_h$. Using the same subset,

Table 6: The importance of data scarcity on model performance for $Q_{le}$ and $Q_h$ estimations.

| Data percent | $Q_{le}$ | | $Q_h$ | |
|---|---|---|---|---|
| | MAE (W/m$^2$)$\downarrow$ | $r\uparrow$ | MAE (W/m$^2$)$\downarrow$ | $r\uparrow$ |
| 20% | 15.2 | 0.4135 | 24.9 | 0.8726 |
| 30% | 13.7 | 0.4268 | 30.2 | 0.8396 |
| 50% | 13.7 | 0.4776 | 27.1 | 0.8979 |

we trained UCformer_ob, a baseline model with the same architecture and hyperparameters but without any simulation-based training. Additionally, CLMU was run over the test period, driven by real-world atmospheric forcing, to provide a physics-based reference.

Fig. 3 presents the results of the three experiments. UCformer (Fig. 3 (a), (d)) significantly outperforms the physics-based model CLMU (Fig. 3 (b), (e)) in estimating $Q_{le}$ and $Q_h$, achieving higher $r$ and lower MAE. Furthermore, the observed-data-only model UCformer_ob (Fig. 3 (c), (f)) performs noticeably worse than UCformer. However, it is observed that neither physics-based nor data-driven models are capable of appropriately estimating negative values of urban heat flux. Despite these challenges, UCformer has demonstrated strong potential to extend to real-world data in addressing the challenge of observational data scarcity.

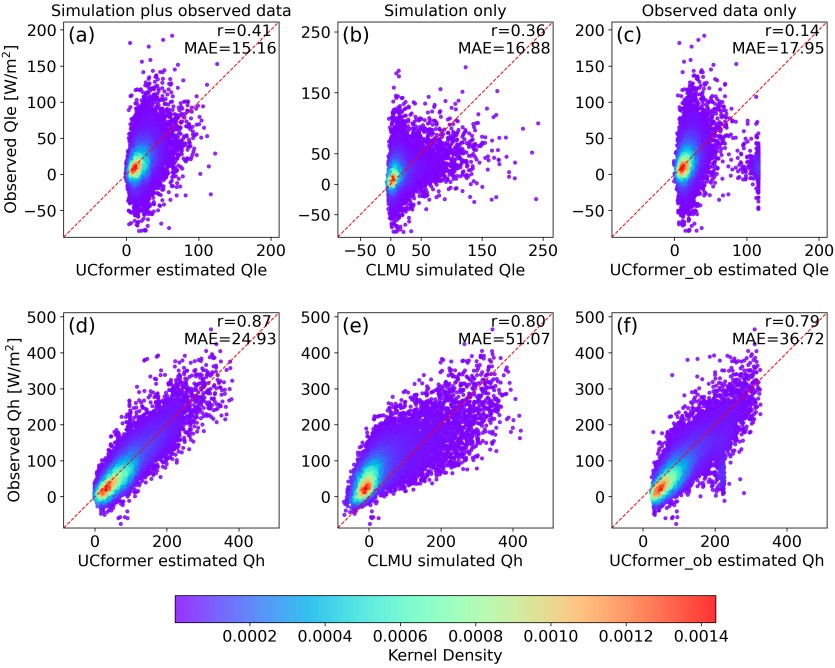

Figure 3: Comparison of latent $Q_{le}$ and sensible $Q_h$ heat flux estimations from UCformer (left column), CLMU (middle column), and UCformer_ob (right column). UCformer fine-tuned with 20% observed data; CLMU driven by real-world forcing; UCformer_ob trained only on limited observations. The rainbow color represents the kernel density.

## 5    Conclusion

We introduce UCformer, a novel approach that integrates urban climatic and physical knowledge to model the interaction between atmospheric forcing and urban surfaces in urban climate systems. Our model outperforms relevant baselines, achieving lower RMSE and demonstrating superior emulation skills, while also exhibiting enhanced generalization to future urban climate dynamics. For the scarcity of real-world data problems, our model showcases strong potential for extension to real-world settings, outperforming a numerical model and a baseline model trained only on observational data in this regard. This work bridges ML techniques with the pressing needs of urban climatology, but it is constrained by limited data availability. Long-term atmospheric forcing and simulated urban climate data are crucial yet scarce, hindering the evaluation of UCformer under diverse climate change scenarios. Additionally, although the models' temporal and spatial generalization has been evaluated, further work is needed to assess their performance across different spatial scales. This represents another limitation of the present study. While ML continues to show promise in climate projections, local urban climate modeling remains critically understudied. We hope this work stimulates the ML community to pursue novel ML approaches tailored to the unique challenges of heterogeneous urban environments.

## Acknowledgements

Zhonghua Zheng appreciates the support provided by the academic start-up funds from the Department of Earth and Environmental Sciences at The University of Manchester.

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

# Appendix

## A    Addtional results and figures

### A.1    Model efficiency

In addition to UCformer, this work evaluated two representative sequence-to-sequence models, that is, LSTNet and Informer, which can be adapted to urban climate estimation.

All models were tested with a batch size of 128 on an NVIDIA 5090 GPU. The inference time for UCformer on our test set (219,000 data points) is approximately 1.4 seconds, with a model size of about 9.35 million parameters. In comparison, Informer requires about 3.1 seconds for inference and has 11.38 million parameters, while LSTNet is more lightweight with 111.13 thousand parameters and an inference time of about 2.3 seconds. The inference efficiency (covering 5-year simulations for 6 cities) of the current model meets the requirements for operational urban climate applications.

These results indicate that, compared to Informer (a transformer-based model), our model has fewer parameters and a shorter inference time. Relative to a smaller LSTNet, which utilizes a CNN+RNN architecture, UCformer also demonstrates an advantage in inference efficiency.

### A.2    Impact of urban representation on local urban climate emulation

Based on the work of Zheng et al. [47], Zhao et al. [46], we exclude urban surface features from the mapping of atmospheric forcing to urban climate to show the impact of urban representation on local urban climate emulations. The results in Fig. 4 are based on the vanilla Transformer, one of the benchmark models used in this work. In this comparison experiment, we maintain consistent hyperparameters, training data, and test data, except for the excluded urban surface features.

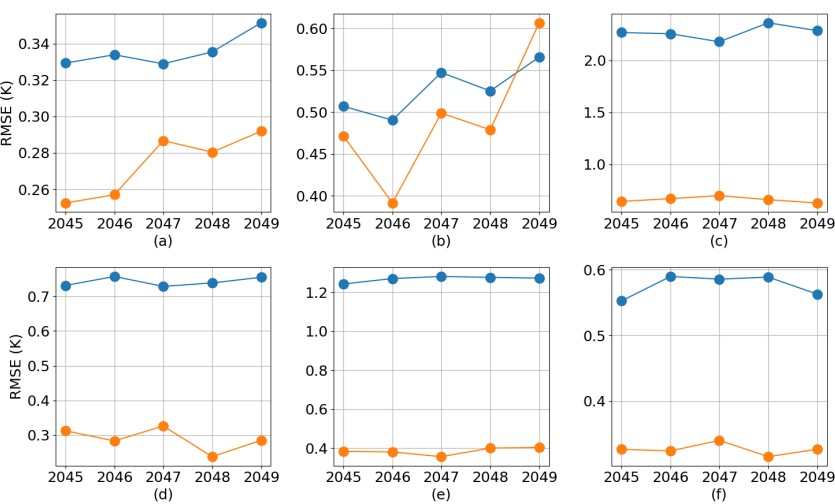

Figure 4: Ablation study regarding the urban representation in urban climate modeling. (a) to (f) denote the emulations of temperature for six cities. Orange dots indicate estimation results with the urban representation included, and blue dots indicate estimation results in the absence of the urban representation.

### A.3    Impact of the interdependency of timesteps on a fine-grained temporal scale

We use the same dataset as well as hyperparameters to reveal the impact of time series on urban climate emulations. The results based on our model are presented in Fig. 5, when interdependencies between time steps are ignored on fine-grained time scales, the emulated representation of the local urban climate is compromised.

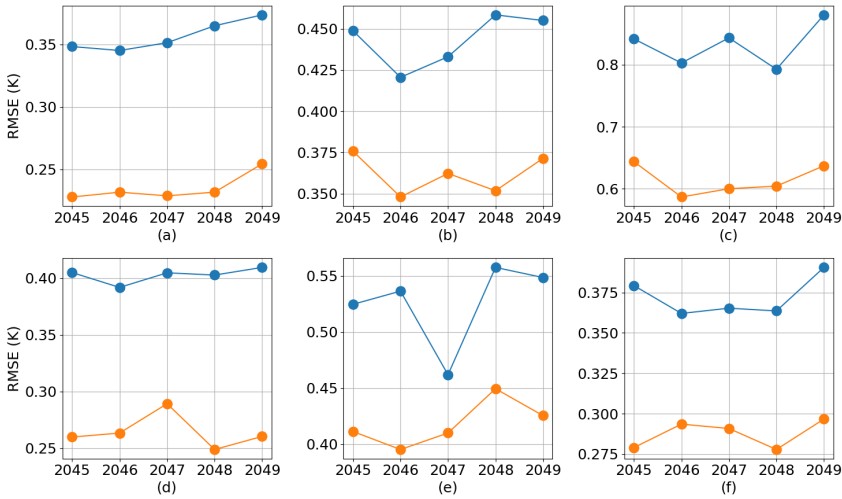

Figure 5: Ablation study regarding the interdependency of timesteps in urban climate modeling. (a) to (f) denote the emulations of temperature for six cities. Orange points indicate that interrelated time series are explicitly characterized in the input feature matrix, while blue points represent time series that are not interrelated within the input feature matrix.

## A.4 The correspondence between the physics-based model elements and the ML modules

Table 7 clarifies the mapping between the model components and the elements of the inference scheme.

Table 7: Mapping between physics-based components and machine learning modules.

| Model Component | Role in Inference Scheme | Input(s) | Output(s) |
|---|---|---|---|
| forcing embedding layer | encodes atmospheric forcing characteristics | atmospheric forcing features | forcing tokens |
| urban surface embedding layer | encodes urban surface feature characteristics | urban surface features | urban surface tokens |
| urban surface-atmosphere interaction block | learns interactions between urban surfaces and atmosphere | forcing query tokens, urban surface key tokens, urban surface value tokens | interaction-encoded tokens |
| surface fluxes interaction block | learns urban surface-surface radiatively interaction | updated interaction-encoded tokens | intermediate features tokens |
| decoder branch 1 | $T$ (temperature) estimation | intermediate $T$ query tokens, concatenated q and t tokens (as keys and values) | esitmated $T$ |
| decoder branch 2 | $q$ (specific humidity) estimation | intermediate $q$ query tokens, concatenated T and t tokens (as keys and values) | esitmated $q$ |
| decoder branch 3 | $t$ (dew point temperature) estimation | intermediate $t$ query tokens, concatenated T and q tokens (as keys and values) | esitmated $t$ |

## A.5 The discrepancy of data distribution

Fig. 6 shows the climatic shift in urban atmospheric conditions over a long-term climate change (from 2044 to 2074) under the SSP3-7.0 scenario, resulting in distributional differences across datasets. The first column (from left) represents atmospheric temperature, the second column shows downwelling shortwave radiation, and the third column shows downwelling longwave radiation.

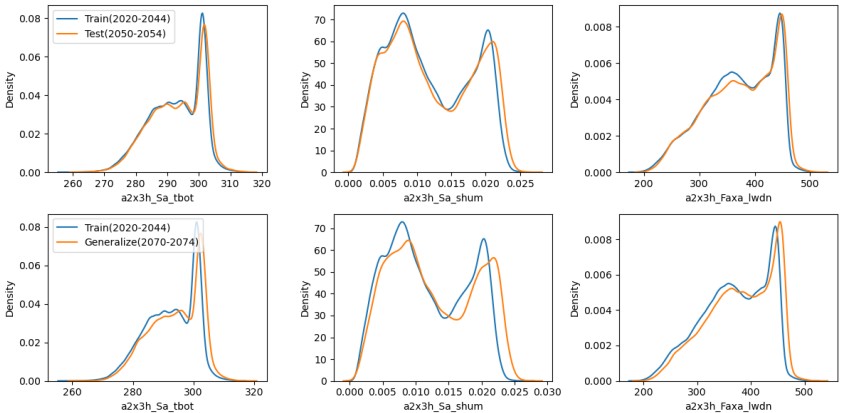

Figure 6: Differences in the data distribution of key atmospheric forcing variables ($X_f$) across datasets. The first column (from left) represents atmospheric temperature, the second column shows downwelling shortwave radiation, and the third column shows downwelling longwave radiation.

## A.6 Statistical significance metric

The paired Wilcoxon signed-rank test is a non-parametric statistical method [35] and was conducted to evaluate the significance of differences between model estimations. Tab. 8 shows the details of Paired Wilcoxon signed-rank tests for model results.

Table 8: The Results of Paired Wilcoxon signed-rank tests. For all models, except for the ablation study (paired test with Transformer results), the results were paired and tested against those of UCformer.

|  | $T$ | $q$ | $t$ |
|---|---|---|---|
| In-distribution | | | |
| AutoML | ** | | ** |
| Transformer | ** | ** | ** |
| MLP_CSB | ** | ** | ** |
| Generalization (Out of Sample) | | | |
| AutoML | ** | ** | ** |
| Transformer | ** | ** | ** |
| MLP_CSB | ** | ** | ** |
| Ablation studies | | | |
| UCformer(DE) | ** | ** | ** |
| UCformer(PD) | ** | ** | ** |
| UCformer_single | ** | ** | ** |
| UCformer(RA) | ** | ** | ** |
| UCformer(TH) | ** | ** | ** |
| UCformer(RA&TH) | ** | ** | ** |

*\*\* represents p <0.05 (95% confidence level), \* indicates p <0.10 (90% confidence level), and non-marked values show no significant difference at the 90% confidence level.*

In the modeling of interactions between urban surfaces and the atmosphere, urban surface parameters are often difficult to acquire and update. Therefore, in this section, we quantified the impact of different urban surface parameters on UCformer emulation performance to explore the importance of these parameters in data-driven urban climate emulation. For this purpose, three different feature datasets, each excluding specific urban surface parameters, were provided to re-train UCformer. In particular, urban surface parameters leveraged in CLMU can be divided into three subsets, the morphological parameters (height-to-width ratio, the fraction of roof, impervious road, etc.), radiative parameters (albedo and emissivity of roof, wall and road) and thermal parameters (heat capacity and thermal conductivity of roof, wall and road) [11]. As such, as shown in Tab. 9, three datasets and

their corresponding models were developed to evaluate the impact of the radiative parameters (RA), the thermal parameters (TH), and the combination of the two parameters, respectively, on UCformer emulation skill.

An insight can be gained in Tab. 9 that $T$ estimation is less affected by the lack of urban radiative parameters but more affected by the absence of thermal parameters, which aligns with the sensitivity study based on physics-based simulations [24]. Overall, the importance of urban surface parameters varies by variable, underscoring the need to tailor urban surface feature selection to improve the emulation accuracy of multiple climate variables.

Table 9: Ablation study regarding the impact of different urban surface sets ($X_s$) on urban climate emulations. RA and TH refer to radiative and thermal parameters, respectively.

| 2050-2054 | $X_s$ | | | $T$ | | $q$ | | $t$ | |
|---|---|---|---|---|---|---|---|---|---|
| method | RA | TH | $\text{MESS}_{agg}\uparrow$ | RMSE (K)$\downarrow$ | MESS$\uparrow$ | RMSE (kg/kg)$\downarrow$ | MESS$\uparrow$ | RMSE (K)$\downarrow$ | MESS$\uparrow$ |
| | ✗ | ✗ | 1.8668 | 0.3495 | 0.8252 | 0.0002 | 0.5428 | 0.2046 | 0.4988 |
| UCformer | ✗ | ✓ | 1.9473 | **0.3211** | **0.8462** | 0.0002 | 0.5505 | 0.1852 | 0.5507 |
| | ✓ | ✗ | 1.9236 | 0.3463 | 0.8276 | 0.0002 | 0.5389 | 0.1891 | 0.5571 |
| | ✓ | ✓ | **2.0468** | 0.3264 | 0.8405 | **0.0002** | **0.6157** | **0.1777** | **0.5906** |

## A.7 Generalize performance

The PDF of the models for the three predictands in 2070-2074 are shown in Fig. 7. UCformer demonstrates a strong generalization ability to approximate all predictands distributions across a time scale. However, AutoML, the Transformer model, and MLR consistently deliver notable deviations of three predictands.

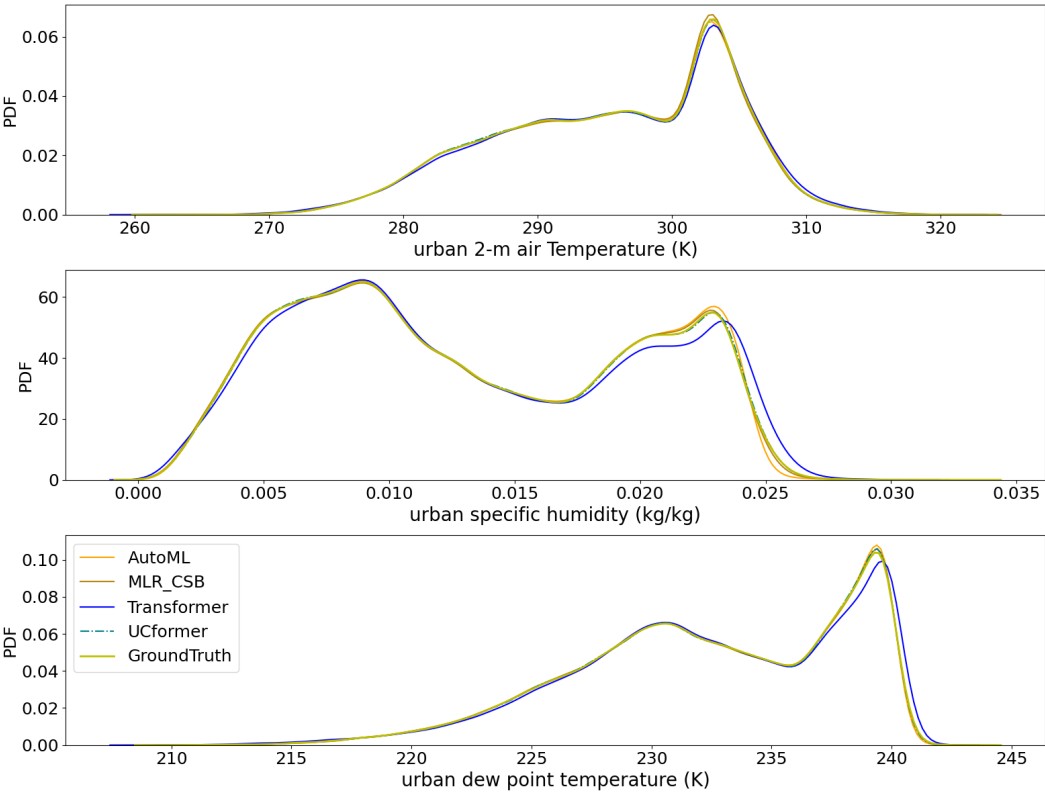

Figure 7: Probability distributions of urban 2-m air temperature (a), specific humidity (b), and dew point temperature (c) for the years 2070-2074. GroudTruth denotes the simulation results of CLMU.

### A.8  Per-city results of shared correlation learning in urban climate emulation

Tab. 10 presents the per-city results under two training settings: shared multi-city training and independent city-specific training. The results highlight consistent performance gains from shared modeling in most cities, with the exception of Singapore.

Table 10: Per-city RMSE results. Values marked with (∘) correspond to city-specific modeling.

| 2050–2054 | $T$ | | | $q$ | | $t$ | |
|---|---|---|---|---|---|---|---|
| method | RMSE(K) | RMSE(K)(∘) | RMSE(g/kg) | RMSE(g/kg)(∘) | RMSE(K) | RMSE(K)(∘) |
| London | **0.2419** | 0.2801 | **0.1117** | 0.1558 | **0.1655** | 0.2198 |
| New York city | **0.3515** | 0.4990 | **0.1645** | 0.2400 | **0.2212** | 0.3253 |
| Shanghai | **0.4437** | 0.5123 | **0.2611** | 0.3968 | **0.2426** | 0.3119 |
| Singapore | 0.2412 | **0.1733** | 0.2130 | **0.1860** | 0.1036 | **0.0872** |
| Sao Paulo | **0.3561** | 0.4427 | **0.2074** | 0.2670 | **0.1435** | 0.2018 |
| Rome | **0.3242** | 0.4573 | **0.1477** | 0.2211 | **0.1896** | 0.2888 |

## B  Implementation details

### B.1  Simulation implementation and urban canyon scheme

Community Land Model (CLM) is an important component of the Community Earth System Model (CESM) [10]. The urban representation/parameterization of CLM (CLMU) was developed by Oleson et al. [26] to simulate the urban energy and flux calculation processes. CLMU is driven by the atmospheric forcing data provided by the atmosphere component of CESM or observational data and is a valuable tool for urban climate study [46, 45, 16, 33, 42, 34]. CLMU adopts an urban canyon scheme to achieve urban surface-atmospheric processes [26]. The urban canyon scheme of CLMU comprises the roof, sunlit and shaded walls, impervious and pervious roads. The parameters of these urban surfaces include morphological characteristics (e.g., height-to-width ratio, roof fraction, pervious road fraction, building height, and roof thickness), radiative properties (e.g., emissivity and albedo of roofs, roads, and walls), and thermal properties (e.g., thermal conductivity and heat capacity of roofs, roads, and walls) [11]. The two-dimensional structure of an urban canyon is characterized by a road at the center, flanked by building walls and rooftops on both sides (with varying height-to-width ratios across different cities). Specifically, the energy balance process in CLMU is subject to urban surface input parameters and is calculated by the flux interactions within a simplified bulk urban air mass. Additionally, it should be noted that the urban energy balance in the current version of CLMU operates independently within a sub-grid [15], enabling a focused investigation of the urban surface-atmospheric processes. We refer readers to the CLMU technique document [25] for the detailed description of the urban canyon scheme. Combined with the exceptional performance of CLMU demonstrated in an urban land surface model comparison project [19], it is well-suited as a benchmark for developing urban climate emulators.

We conducted CLMU simulations under the SSP3-7.0 scenario for six cities (London, New York City, Shanghai, Singapore, Sao Paulo, and Rome) with diverse background climates and urban surfaces to obtain training, validation, and testing data for data-driven methods. To be specific, CLMU was conducted to output instantaneous urban climate simulation outcomes for 8 time steps per day (every 3 hours) for the period of 2020 to 2074 after a 5-y spin-up (2015-2019). The simulation data were divided into 4 subsets: data for the period 2020–2044 were used as training data (1,095,000 data points), 2045–2049 data were used for validation (219,000 data points), and data from 2050–2054 (219,000 data points) were mainly used to evaluate model performance. Besides, we further used simulation data from 2070–2074 (219,000 data points) to test model generalization ability.

### B.2  Atmospheric forcing and urban surfaces variables

This work screened the forcing data from the atmospheric component of the Community Earth System Model (CESM) at a native resolution of $0.9° \times 1.25°$. Urban surface characteristics were extracted from CLMU's static surface input files at the same resolution. The CLMU output in single-point mode shares this spatial resolution, which aligns with the default resolution used in global urban climate projections. However, it is worth noting that the objective of this work is to capture the underlying physical mechanisms governing urban climate, rather than to perform estimations tied to a specific spatial resolution. This design choice inherits the resolution-agnostic nature of the

physics-based model. CLMU can operate in a single-point mode and does not assume a fixed spatial resolution. Instead, its spatial footprint is implicitly determined by the resolution of the atmospheric forcing and surface input data. As demonstrated in Section 4.6, both our model and CLMU are able to operate using flux tower data with a finer spatial granularity (~500 m), further supporting the resolution-agnostic nature of the learned physical processes.

The variables used to develop the data-driven models are listed in Table 11. To enhance training efficiency (except for the decision tree-based model), all data were first standardized using the mean and standard deviation of the training set. After standardization, the training data were scaled to the range (-1, 1) based on the minimum and maximum values of the standardized training set [18, 4].

Table 11: List of features considered in this study. all fields are taken from the CLM simulation. features marked with * were omitted in the model development due to they have constant or nearly identical values across the training area.

| Type | Fields | Description | Unit |
|---|---|---|---|
| Embedding as atmospheric forcing | a2x3h_Faxa_swndr | Direct near-infrared incident solar radiation | $\text{W m}^{-2}$ |
| | a2x3h_Faxa_swvdr | Direct visible incident solar radiation | $\text{W m}^{-2}$ |
| | a2x3h_Faxa_swndf | Diffuse near-infrared incident solar radiation | $\text{W m}^{-2}$ |
| | a2x3h_Faxa_swvdf | Diffuse visible incident solar radiation | $\text{W m}^{-2}$ |
| | a2x3h_Faxa_rainc | Convective precipitation rate | $\text{kg m}^{-2}\text{s}^{-1}$ |
| | a2x3h_Faxa_rainl | Large-scale (stable) precipitation rate | $\text{kg m}^{-2}\text{s}^{-1}$ |
| | a2x3h_Faxa_snowc | Convective snow rate (water equivalent) | $\text{kg m}^{-2}\text{s}^{-1}$ |
| | a2x3h_Faxa_snowl | Large-scale (stable) snow rate (water equivalent) | $\text{kg m}^{-2}\text{s}^{-1}$ |
| | a2x3h_Sa_u | Zonal wind at the lowest model level | $\text{m s}^{-1}$ |
| | a2x3h_Sa_v | Meridional wind at the lowest model level | $\text{m s}^{-1}$ |
| | a2x3h_Sa_tbot | Temperature at the lowest model level | K |
| | a2x3h_Sa_shum | Specific humidity at the lowest model level | $\text{kg kg}^{-1}$ |
| | a2x3h_Sa_pbot | Pressure at the lowest model level | Pa |
| | a2x3h_Faxa_lwdn | Downward longwave heat flux | $\text{W m}^{-2}$ |
| | a2x3h_Sa_z | Height at the lowest model level | m |
| | doma_lon | Longitude | deg |
| | dom_lat | Latitude | deg |
| Embedding as urban surface features | CANYON_HWR | Canyon height to width ratio | |
| | WTLUNIT_ROOF | Fraction of roof | unitless |
| | WTROAD_PERV | Fraction of pervious road | unitless |
| | EM_IMPROAD* | Emissivity of impervious road | unitless |
| | EM_PERROAD* | Emissivity of pervious road | unitless |
| | EM_ROOF | Emissivity of roof | unitless |
| | EM_WALL | Emissivity of wall | unitless |
| | ALB_IMPROAD_DIR* | Direct albedo of impervious road | unitless |
| | ALB_ROOF_DIR | Direct albedo of roof | unitless |
| | ALB_PERROAD_DIR* | Direct albedo of pervious road | unitless |
| | ALB_WALL_DIR | Direct albedo of wall | unitless |
| | TK_ROOF | Thermal conductivity of roof | $\text{W m}^{-1}\text{K}^{-1}$ |
| | TK_WALL | Thermal conductivity of wall | $\text{W m}^{-1}\text{K}^{-1}$ |
| | TK_IMPROAD | Thermal conductivity of impervious road | $\text{W m}^{-1}\text{K}^{-1}$ |
| | CV_ROOF | Volumetric heat capacity of roof | $\text{J m}^{-3}\text{K}^{-1}$ |
| | CV_WALL | Volumetric heat capacity of wall | $\text{J m}^{-3}\text{K}^{-1}$ |
| | CV_IMPROAD | Volumetric heat capacity of impervious road | $\text{J m}^{-3}\text{K}^{-1}$ |
| | COSZEN | Cosine of solar zenith angle | deg |

## B.3 Model configuration

Table 12: Hyperparameters used to train UCformer are listed along with their optimal values and the search space for tuning the unconstrained model, shown as the range or set of possible values and the sampling method. Note that attention heads and hidden size were not included in the optimization process due to the structural requirements of UCformer.

| Hyperparameter | Value | Search |
|---|---|---|
| layers | 4 | [3-8] $\sim$ choice |
| feed-forward dimension | 2048 | 512, 1024, 2048 $\sim$ choice |
| batch size | 128 | 32, 64, 128 $\sim$ choice |
| learning rate | 1e-5 | [1e-5, 1e-2] $\sim$ log uniform |
| dropout rate | 0.1 | [0.1-0.4] $\sim$ choice |
| epochs | 50 | [50-200] $\sim$ choice |
| attention heads | 3 | |
| hidden size | 384 | |

Table 13: Hyperparameters used to train Transformer are listed along with their optimal values and the search space for tuning the unconstrained model, shown as the range or set of possible values and the sampling method.

| Hyperparameter | Value | Search |
|---|---|---|
| layers | 8 | [3-8] $\sim$ choice |
| feed-forward dimension | 2048 | 512, 1024, 2048 $\sim$ choice |
| batch size | 32 | 32, 64, 128 $\sim$ choice |
| learning rate | 1e-5 | [1e-5, 1e-2] $\sim$ log uniform |
| dropout rate | 0.14 | [0.1-0.4] $\sim$ choice |
| epochs | 100 | [50-200] $\sim$ choice |
| attention heads | 4 | 2,4,8 $\sim$ choice |
| hidden size | 128 | 128, 256, 512 $\sim$ choice |

The configuration of UCformer and Transformer is listed in Tabel 12 and Tabel 13.

For AutoML, we used a Python package named "FLAML" [37] to achieve automated model selection and hyperparameter tuning (Automated Machine Learning, AutoML) for decision tree-based models. Specifically, FLAML enables automated optimal machine learning model selection and optimizes the search processes based on the evaluation metric and computational efficiency, that is, the "time budget" hyperparameter. The package then allows iterative selection of learners, hyperparameters, sample sizes, and resampling strategies. In this work, we configured AutoML for a regression task with an "auto" estimator list, optimizing the RMSE metric, and set a time budget of 3,600 seconds (1 hour) per trial. The "auto" estimator list includes tree-based models such as LightGBM [13], XGBoost[8], and Random Forest.

## B.4 Fine-tuning model with real-world data

To align with the observational dataset, we modified the input features of UCformer (from 30 to 23) and the output features (from 3 to 2), while keeping all parameters of UCformer updated during the fine-tuning process. The new predictions of sensible and latent heat fluxes are constrained by the surface energy balance. This makes the physics-guided decoder in UCformer naturally adaptable to predicting these variables.

## C Code of Ethics and Broader Impacts

Our research is ethical. The physics-guided model proposed in this paper enables local urban climate emulation, offering valuable applications in fields such as climate-informed urban planning and providing significant benefits to society.

The dataset used in this study is derived from a public numerical model, ensuring no issues related to infringement or privacy leakage. The experiments conducted are fair, reproducible, and designed

with minimal resource consumption, posing no environmental or societal impact. Furthermore, the proposed model is free from bias and discrimination concerns. To promote transparency and accessibility, we open-source the model code and checkpoints on GitHub.

## D  Safeguard of Model

This paper introduces a physics-guided model for local urban climate emulation. It is essential to recognize that all models inherently contain some level of emulation error. Therefore, the proposed model should not be used as the sole basis for predicting major events or policymaking. Instead, its findings should be combined with other models and expert knowledge to ensure more comprehensive and well-informed conclusions.

## E  Assets

Our study complies with the licensing requirements for existing assets, as the data used in this paper are generated from an open-source model that is publicly accessible and authorized for academic research. The model proposed in this study constitutes a novel contribution and is regarded as a new asset.

