# OpenReview forum: "Learning Urban Climate Dynamics via Physics-Guided Urban Surface–Atmosphere Interactions"
_NeurIPS.cc/2025/Conference — NeurIPS 2025 poster_

### Official Review · Reviewer_whR3 · 2025-06-29

**Clarity:** 3
**Significance:** 3
**Originality:** 4
**Rating:** 5
**Confidence:** 3

**Summary:**

The authors propose an attention-based approach that integrates urban climate knowledge and physical models to capture interactions between atmospheric phenomena and urban surfaces. Their method utilizes an encoder to model atmospheric–surface interactions and a decoder to guide the physical models. The experiments cover dataset descriptions, baseline comparisons, and ablation studies aiming to assess the robustness of the approach.

**Questions:**

1. I suggest adding a table that clarifies the mapping between the model and the elements of the inference scheme.
2- I suggest providing a minimal code example that loads the data and runs the test exercises.
3. Evaluate with a held-out city to demonstrate spatial robustness.

**Ethical Concerns:**

["NO or VERY MINOR ethics concerns only"]

**Final Justification:**

Thanks for the opportunity of reviewing this article. I appreciate the thoughtful insights provided by the reviewers and the arguments presented by the authors. After careful consideration, I decided to keep my score. Thanks

**Limitations:**

Yes

**Paper Formatting Concerns:**

No major issues but consider the following typos:


*** several in line 168
*** Table in line 555

**Quality:**

4

**Strengths And Weaknesses:**

The authors present an approach that combines data with physical models to study climatological phenomena in urban environments. Their results are outstanding, and ablation studies confirm that the correct components were integrated. They also provide code to reproduce the findings.

However, there should be more emphasis on clarifying how the physical models connect to the machine-learning models, and the code should include additional details to make it easier to use.

---

> ### Author Rebuttal · Authors · 2025-07-30
>
> **Q1: More emphasis on clarifying how the physical models connect to the machine-learning models**.
>
> **Response:**
>
> We thank the reviewer for this valuable suggestion. Following your advice, we have added a table to clarify the mapping between the model components and the elements of the inference scheme. This table will be included in the appendix of the revised manuscript.
>
> | Model Component                            | Role in Inference Scheme                                  | Input(s)                                                     | Output(s)                    |
> | ------------------------------------------ | --------------------------------------------------------- | ------------------------------------------------------------ | ---------------------------- |
> | forcing embedding layer                    | encodes atmospheric forcing characteristics               | atmospheric forcing features                                 | forcing tokens               |
> | urban surface embedding layer              | encodes urban surface feature characteristics             | urban surface features                                       | urban surface tokens         |
> | urban surface-atmosphere interaction block | learns interactions between urban surfaces and atmosphere | forcing query tokens, urban surface key tokens, urban surface value tokens | interaction-encoded tokens   |
> | surface fluxes interaction block           | learns urban surface-surface radiatively interaction      | updated interaction-encoded tokens                           | intermediate features tokens |
> | decoder branch 1                           | *T* (temperature) estimation                              | intermediate *T* query tokens, concatenated *q* and *t* tokens (as keys and values) | esitmated *T*                |
> | decoder branch 2                           | *q* (specific humidity) estimation                        | intermediate *q* query tokens, concatenated *T* and *t* tokens (as keys and values) | esitmated *q*                |
> | decoder branch 3                           | *t* (dew point temperature) estimation                    | intermediate *t* query tokens, concatenated *T* and *q* tokens (as keys and values) | esitmated *t*                |
>
>
>
> **Q2: The code should include additional details to make it easier to use.**
>
> **Response:**
>
> We appreciate the reviewer’s suggestion, which made us realize the importance of providing an easy-to-use example for model training and inference. Although we have aimed to ensure full transparency regarding our technical design, model structure, loss functions, and experimental setup, we acknowledge that we have not included the data loading or training scripts in the current code release during the review process. We will release additional code, including data loading and training routines, as well as a simple example for model training and inference, to facilitate reproducibility.
>
>
>
> **Q3: Evaluate with a held-out city to demonstrate spatial robustness.**
>
> **Response:**
>
> Spatial generalization remains quite a challenging issue in global climate and urban climate prediction/estimation. While our work primarily focuses on improving temporal generalization, we agree that evaluating spatial generalization is crucial for real-world applications. To provide further insight, we conducted additional “leave-one-out” experiments. When selecting training cities, we intentionally chose cities with significantly different urban morphology or climate regimes. Therefore, to better assess spatial generalization, we performed supplementary experiments where one city was held out during training. The results are provided below for your reference (results in parentheses indicate performance under the original setting).
>
> | 2050-2054     |                 | T               |                 | q               |                 | t               |                 |
> | ------------- | --------------- | --------------- | --------------- | --------------- | --------------- | --------------- | --------------- |
> | Held-out city | MESSagg         | RMSE(K)         | MESS            | RMSE(g/kg)      | MESS            | RMSE(K)         | MESS            |
> | London        | 1.6629 (1.9007) | 0.2990 (0.2419) | 0.8127 (0.8541) | 0.1344 (0.1117) | 0.4357 (0.5171) | 0.1977 (0.1655) | 0.4144 (0.5295) |
> | New York City | 1.6989 (1.8901) | 0.4167 (0.3515) | 0.7699 (0.8030) | 0.1904 (0.1645) | 0.5063 (0.5741) | 0.2507 (0.2212) | 0.4225 (0.5130) |
> | Shanghai      | 1.9369 (2.0024) | 0.4806 (0.4437) | 0.8349 (0.8486) | 0.2886 (0.2611) | 0.5555 (0.5986) | 0.2529 (0.2426) | 0.5465 (0.5552) |
> | Singapore     | 2.2840 (2.4586) | 0.2668 (0.2412) | 0.8581 (0.8727) | 0.2798 (0.2130) | 0.7060 (0.7975) | 0.1279 (0.1036) | 0.7198 (0.7884) |
> | Sao Paulo     | 1.8953 (2.1241) | 0.3965 (0.3561) | 0.8179 (0.8363) | 0.2587 (0.2074) | 0.5326 (0.6618) | 0.1649 (0.1435) | 0.5448 (0.6260) |
> | Rome          | 1.8163 (1.9048) | 0.3972 (0.3242) | 0.7843 (0.8282) | 0.1571 (0.1477) | 0.5197 (0.5450) | 0.1975 (0.1896) | 0.5123 (0.5316) |
>
> The results show that UCformer exhibits a certain degree of spatial generalization, but its performance varies notably across cities and is strongly influenced by both urban morphology and local climate conditions. For instance, the RMSE of temperature estimates ranges from approximately 0.02 K in Singapore to 0.07 K in Rome, while the RMSE of dew point temperature estimates ranges from about 0.01 K in Rome to 0.03 K in London.
>
> Overall, these findings suggest that while UCformer is capable of spatial generalization, its performance is influenced by the uniqueness of the target city’s urban and climatic conditions. We acknowledge spatial generalization as an important direction for future research. In particular, we plan to further investigate the shared representations of urban climate processes proposed in Section 4.5, and aim to explore whether scaling laws emerge with respect to the number and diversity of cities included during training, and how these factors affect the model's spatial generalization capability across heterogeneous urban environments and climate regimes.

---

### Official Review · Reviewer_CyZa · 2025-07-01

**Clarity:** 2
**Significance:** 3
**Originality:** 2
**Rating:** 3
**Confidence:** 4

**Summary:**

This paper proposes UCformer, a Transformer-based architecture that integrates surface–atmosphere cross-attention and soft physics constraints for urban climate variables. It also introduces a new dataset built from multi-city, multi-decadal CLMU simulations. UCformer demonstrates improved RMSE and emulation skill over simple MLP, AutoML tree ensembles, and a vanilla Transformer.

**Questions:**

See weakness.

**Ethical Concerns:**

["NO or VERY MINOR ethics concerns only"]

**Final Justification:**

The authors have addressed the first two points of my concerns. But there are still lingering problems as suggested in my response.

**Quality:**

1

**Strengths And Weaknesses:**

Strengths
S1. Novel and important downstream tasks. Predicting urban climate variables at 3-hour resolution—2 m air temperature, specific humidity, and dew point—is both timely and practically valuable.
S2. Physics-guided design. The cross-attention encoder and decoder inject domain knowledge softly, offering interpretability without hard constraints.
S3. A new dataset. The CLMU simulation corpus (2020–2074) across six cities with chronological train/val/test/generalization splits enables robust evaluation.

Weakness
While the paper’s motivation and contributions are clear, it overlooks a vast body of multivariate time-series forecasting research, leading to two critical issues:

W1. Problem definition ignored.
The authors’ own formulation (Sec. 3.1) makes it explicit:
– Inputs: $X_f\in\mathbb{R}^{T\times C\times N}$, a sequence of $C$ forcing variables over $T$ timesteps across $N$ cities
– Outputs: multi-step predictions of temperature, humidity, dew point over the same window
This is precisely a multivariate sequence-to-sequence forecasting task. Yet the manuscript fails to cite or discuss decades of work on RNN/LSTM forecasters, TCNs/CNNs for sequences, advanced Transformers for time series (Informer, Temporal Fusion Transformer, Autoformer, PatchTST), or even classical VAR/ARIMA methods.

W2. Inadequate baselines.
The experimental section only benchmarks UCformer against:

1. An MLP “ClimSim-style” regression
2. An AutoML-selected tree ensemble
3. A vanilla Transformer with no temporal forecasting enhancements

It omits specialized forecasting models such as LSTNet and DeepAR, TCN-based architectures, Informer/PatchTST, and VAR/ARIMA. Without these comparisons, we cannot determine whether UCformer’s gains derive from its physics guidance or simply from its choice of baselines unsuitable for time series.

W3. Implementation.
Beyond the aforementioned problems, the core technical part of the paper is essentially not reproducible. Until the review time (6.30 AOE), the authors haven't fully released the code in their provided anonymous GitHub link. More importantly, 1) none of the novel physics components discussed in Section 3.4 appears in the provided code, 2) in the main implementation, no non-standard architecture appears other than naming things “climate_attention” and splitting the decoder into three branches. Everything is built on vanilla PyTorch Transformer primitives, significantly different from the paper's description, and 3) No data loading, loss functions, or training code is provided. Hence, it is impossible to verify the true technical designs of the proposed framework.

---

> ### Author Rebuttal · Authors · 2025-07-30
>
> **Q1: Problem definition.**
>
> **Response:**
>
> We have to clarify that time-series forecasting is fundamentally different from the problem we address in this work. In forecasting, the model typically receives a sequence of past target labels (e.g., temperature at previous time steps) as part of the input in order to predict future values. In contrast, our task does not include any past target labels in the input. Instead, we aim to estimate a sequence of urban climate variables (e.g., temperature, humidity) based solely on external predictors, namely, atmospheric forcing and urban surface characteristics within the same time window, rather than predicting future values based on past observations.
>
> As noted in Strength S1, the reviewer acknowledges that our work proposes a novel and important downstream task, which centers on the significant divergence between urban and background climates. Specifically, our goal is to explore how urban climates are shaped by atmospheric forcing and urban surface characteristics at a given time. For this reason, the models mentioned by the reviewer, such as VAR, ARIMA, PatchTST, and Autoformer, are fundamentally incompatible with our task formulation. These models rely heavily on past target labels and their temporal trends to forecast future values. In contrast, our model does not receive any historical target labels as input. Including such information would provide additional signals that could artificially enhance model performance by revealing patterns not accessible in our intended setting. This would compromise the goal of estimating urban climate conditions strictly from external drivers such as atmospheric forcing and surface characteristics. Therefore, these forecasting methods are not directly applicable or comparable in our case.
>
> **Q2: Why not include time-series forecasting baselines.**
>
> **Response:**
>
> As clarified in our Response to Q1, our task is not a conventional time-series forecasting problem. Therefore, when selecting baseline models, we focused on approaches relevant to urban climate estimation or sequence-to-sequence mapping, rather than forecasting. The scarcity of research specifically on urban climate mapping has limited the available choices for suitable baselines, which may also have contributed to the reviewer's misunderstanding of our task as traditional sequence prediction.
>
> Nonetheless, following the reviewer’s suggestions, we additionally evaluated two sequence models that can be reasonably adapted to our framework. We modified their data interfaces accordingly to accept atmospheric forcing and urban surface features as inputs, enabling them to estimate urban climate variables within our task setting.  Specifically, we compared UCformer with Informer_modified and LSTNet_modified, as well as with a multivariate linear regression (MLR) model. It is important to note that, since our task is not time-series forecasting, classical forecasting models such as VAR/ARIMA are not appropriate baselines. Instead, MLR is a more suitable and basic baseline for mapping tasks such as ours.  The supplementary results are provided below:
>
> | 2050-2054         |            | T          |            | q          |            | t          |            |
> | ----------------- | ---------- | ---------- | ---------- | ---------- | ---------- | ---------- | ---------- |
> | Model             | MESSagg    | RMSE(K)    | MESS       | RMSE(g/kg) | MESS       | RMSE(K)    | MESS       |
> | MLR               | 0.2069     | 0.7774     | 0.6112     | 0.3562     | 0.2516     | 0.6551     | -0.6559    |
> | LSTNet_modified   | 1.9065     | 0.3528     | 0.8286     | 0.2238     | 0.5075     | 0.1910     | 0.5704     |
> | Informer_modified | 1.7700     | 0.3567     | 0.8268     | 0.2325     | 0.4775     | 0.2003     | 0.4657     |
> | UCformer          | **2.0468** | **0.3264** | **0.8405** | **0.1851** | **0.6157** | **0.1777** | **0.5906** |
>
> As shown in Table, UCformer achieves the best overall performance across all evaluation metrics compared to MLR, LSTNet_modified, and Informer_modified. These results demonstrate the advantage of our approach for urban climate estimation. We appreciate the reviewer’s suggestion, which allowed us to revisit the suitability of sequence forecasting models as baselines for our sequence regression task.
>
> **Q3: Model implementation  and the designed physical components' codes.**
>
> **Response:**
>
> **1. Model implementation and reproducibility:**
>
> We respectfully disagree with the reviewer’s statement regarding reproducibility. We have fully released the code necessary to reproduce the results presented in the paper. In the manuscript, we clearly describe the data preprocessing steps (Appendix A.2), model training settings (Appendix A.3 and also in the GitHub repository), loss function (Section 4.1 "Training Objective"), and model checkpoint (GitHub repository). We have also made the datasets used in this work available for review. We have aimed for full transparency regarding our technical design, model structure, loss functions, and experimental setup. We believe these materials are sufficient for reproducing our results.
>
> **2. Technical part codes:**
>
> We thank the reviewer for examining our code and for these detailed comments. However, our architectural design deviates meaningfully from vanilla Transformers. Our contribution lies in **adapting the architecture to encode physical process-specific inductive biases**, rather than proposing entirely new operations. This is in line with contemporary scientific machine learning practices, which favor physically meaningful design over novelty for novelty’s sake.
>
> **(1) Regarding the Decoder components described in Section 3.4:**
>
> Section 3.4 and Equations (8)–(12) of our manuscript provide the theoretical basis for the soft physics-inspired constraints in our decoder design, as well as the rationale for using cross-attention to incorporate information from other variables when estimating each target variable. We have consistently stated that we do not embed explicit physical equations as hard constraints in the model. Instead, we aim to represent physically plausible dependencies between target variables through learnable, flexible mechanisms. In addition, Figure 2 (Section 3.2) clearly illustrates the use of cross-attention in the decoding process, which aligns with the reviewer's observation of "splitting the decoder into three branches.". As shown in the code (lines 214–224 of the `forward `function), our implementation of target variable decoding is consistent with both the manuscript and Figure 2: when estimating any given variable, information from the other two variables is also utilized.
>
> **(2) Regarding the Encoder components:**
>
> As shown in Figure 2 and Section 3.2 of the manuscript, our encoder is not a vanilla stack of Transformer blocks, but a multi-stage architecture designed to reflect the layered physical processes in urban climate systems:
>
> The first encoder block uses cross-attention to model interactions between atmospheric forcing and static urban surface features. This is implemented in the `climate_attention` class (code line 46), where `forcing_data` attends to `surface_data` to simulate the modulation of dynamic inputs by static land characteristics.
>
> The second block in the encoder is intended to learn radiative interactions among urban surfaces after atmospheric forcing, as represented by self-attention in Figure 2. These are implemented in the `TransformerEncode` class (code line 131), where the first encoder layer handles atmospheric-surface interactions and subsequent layers capture the updated properties and refine the representation, following the interaction logic described in the paper (code lines 139–140).
>
> **(3) Data loading, loss function, and training code:**
>
> The manuscript clearly presents the data input  (Appendix A.1) and preprocessing steps (Appendix A.2), model training settings (Appendix A.3 and in the GitHub repository), and loss function (Section 4.1 "Training Objective"). The datasets used in this study have also been made available for review.
>
> While the current code release does not include the dataloader and training loop, these components follow standard PyTorch implementations without any task-specific customization. Their omission does not hinder the understanding or verification of the framework’s core technical contributions, which are fully described and reproducible based on the provided materials.
>
> To further support reproducibility, we will include the complete training pipeline—covering data loading and a training/inference example script—in the final code release.
>
> In summary, although we adopt Transformer primitives for transparency and compatibility, **our architectural choices are explicitly designed to reflect domain and physical knowledge**—specifically, the **hierarchical modeling of physical processes** such as large-scale forcing interactions and localized radiative responses. This design goes beyond the standard use of self-attention and presents a **structured inductive bias** rooted in urban climate system behavior. Therefore, we respectfully argue that the UCformer encoder embodies architectural novelty that is both principled and practically motivated.

---

> > ### Comment · Reviewer_CyZa · 2025-08-06
> >
> > Thanks for your detailed rebuttal. My concerns were partially addressed and I would like to reconsider my rating properly. However, some concerns remain as flows:
> >
> > - For the W1, I do understand that your task is not an auto-regressive prediction, where your problem setting is not to use historical labeled data to predict the future (you use other variables to do prediction). However, the input themselves are exactly time series and you do utilize all the information from input (including the history). In such a context, many multivariate time series forecasting models can be directly applied to your data by just changing the output (even without touching their architectures). Hence, it's unconvincing to ignore time series forecasting research (or more broadly, time series analysis) in the manuscripts.
> > - For the W2, I appreciate your great effort in testing the additional baseline. From the new results, it is obvious that LSTNet_modified is better than Informer or Transformer in your original paper. This suggest that time series models may have advantages and should be considered as strong baselines.
> > - For the W3,
> >   - On the reproducibility problem, I do notice that you have released many implementation details in the Appendix and had been reading through them carefully before the rebuttal period. What I'm really very confused is that the code you released is too few. I have carefully checked your anonymous GitHub, which haven't been updated since May 11th. The repo only contains 3 scripts defining model architecture and 1 config files.  Additionally, your huggingface repo only contains datasets. As there lacks environment settings / data processing / inference / evaluation in the code, it will be extremely difficult, if not impossible, to reproduce your results during the reviewing period. Given that these datasets are not widely verified by the NIPS community, I think it is absolutely necessary to provide the complete inference code for your model (and if with one baseline, e.g., the second best MLP_CSB, would be better), so that the readers can access the quality of the datasets and the difficulty of your tasks.
> >   - On the model component problem, while you claim your decoder’s cross‐attention mechanism as “physics‐guided,” I remain unconvinced that this naming is justified. In section 3.4 you listed the physics equations as a background but stated it is not usable due to the lack of pressure. The cross attention is proposed as a substitution. However, the attention merely learns to weight and combine latent representations; it does not enforce energy conservation, mass balance, or any other governing physical equations in a strict sense. The outcomes are purely data-driven, where attention weights are learned from statistical associations in the training data, rather than guided by physical coefficients or conservation laws. To summary, I was still not convinced by the claim that the proposed model architecture are physics-guided.

---

> > > ### Author Response · Authors · 2025-08-08
> > >
> > > Thank you for your thoughtful reply. We appreciate the opportunity to address your remaining concerns and are happy to provide further clarification below:
> > >
> > > **W1:**
> > >
> > > We would like to clarify that our task is fundamentally different from **multivariate time series forecasting (MTSF)**. MTSF models multiple correlated time series by using the historical labels of all related variables to capture their temporal trends, dependencies, and interactions, and then predicts the future values at the next time step. In our case, the inputs are not multiple correlated time series but include variables such as static urban surface parameters, and the target outputs are not the immediate future labels of any time series. Therefore, directly applying MTSF models would fundamentally alter the problem formulation and diverge from the core objective of our work.
> > >
> > > Simply “changing the output” of MTSF models to match our task does not resolve the mismatch in problem formulation. It would impose unnecessary complexity, introduce assumptions about temporal dependence that are not aligned with our objective, and risk misinterpreting the nature of the task.
> > >
> > > **W2:**
> > >
> > > In the submission, we compared three baselines (i.e., MLP_CSB, Transformer, and AutoML) with our proposed model, **UCformer**. In the rebuttal phase, we further included three new baselines: Informer_modified, LSTNet_modified, and MLR.
> > >
> > > As observed, the newly added LSTNet_modified outperforms some of the original baselines, but it remains less effective than the MLP_CSB baseline. Most importantly, **UCformer** consistently outperforms both the original and newly added baselines, demonstrating its robustness and effectiveness. Therefore, in our case time-series models do not constitute stronger baselines than the approaches already included.
> > >
> > > **W3:**
> > >
> > > **(1) Model implementation:** We sincerely thank the reviewer for noting the implementation details in our manuscript and the constructive suggestion. We agree that releasing the complete inference code is important for reproducibility. We commit to providing the full inference code for our model and the baseline to facilitate dataset verification and enable readers to better assess our work
> > >
> > > **(2) The justification of physics-guided ML:** We respectfully disagree with the statement that physics-guided ML must “*strictly enforce energy conservation, mass balance, or other governing equations*”. First, the physical background used in our work (e.g., the empirical Magnus–Tetens formula) is not a universally proven, immutable physical law. Therefore, there is no compelling reason to embed it in a strict, hard-constrained form within the ML architecture. Instead, these physical backgrounds do demonstrate that certain urban climate variables are physically related, and leveraging such relationships to guide the model’s information flow and capture meaningful interactions is both practical and impactful. This approach aligns with current trends in scientific ML [1], where physics-guided ML incorporates physical or domain-specific prior knowledge into models rather than relying solely on hard physical constraints.
> > >
> > > For example, Wang et al.[2] designed TF-Net based on insights from physical modeling and turbulence theory, without incorporating explicit equations. Prior knowledge drives a multilevel decomposition of turbulent flow into intermediate variables, each decomposed and encoded by trainable networks. A shared decoder models their interactions to predict the flow. Prior knowledge thus defines and constrains the latent variables and information flow.
> > >
> > > Additionally, we respectfully disagree with the reviewer's statement that "*The outcomes are purely data-driven, where attention weights are learned from statistical associations in the training data, rather than guided by physical coefficients or conservation laws.*" In our model, attention weights are not learned arbitrarily from data; Instead, they are guided by physical prior knowledge that defines and constrains the flow of information within the network, for example, by enforcing interactions with physically meaningful latent representations (i.e., dew point temperature and specific humidity). This ensures that the learned weights reflect physically meaningful processes, rather than just arbitrary statistical patterns in the data, a design philosophy also exemplified in other domain studies [2].
> > >
> > > In summary, our use of “physics-guided ML” is well-justified and consistent with both established definitions in the literature and recent successful applications in other domains.
> > >
> > > References:
> > >
> > > [1] Wang R, Yu R. Physics-guided deep learning for dynamical systems: A survey[J]. arXiv preprint arXiv:2107.01272, 2021.
> > >
> > > [2] Wang R, Kashinath K, Mustafa M, et al. Towards physics-informed deep learning for turbulent flow prediction[C]//Proceedings of the 26th ACM SIGKDD international conference on knowledge discovery & data mining. 2020: 1457-1466.

---

> ### Comment · Reviewer_CyZa · 2025-08-09
>
> Thanks for the reply. While W2 has been addressed, my concerns on the other points remain. I still feel that the response seems more to dismiss my concerns rather than providing solid evidence from the implementation or mathematics.
>
> **W1:** While you insist on the conceptual difference in your paper, in your implementation, it's not the case. It is evident that all models accept simple sequences with a shape of `[batch_size, seq_length, feature_dim]`. From your own code, the vanilla transformer is applied to your data without changing the model structure or applying any attention mask or constraints. Hence, it is unconvincing to claim "your task is fundamentally different". Your idea may be different, but your baseline implementation is very simple and has no difference from a common processing procedure.
>
> **W3_1: ** Given the particularly few lines of code in the repository and the two very simple baselines you provided, I can't tell the real strength of your method.
>
> For the convenience of other reviewers and the AC, I would like to ground my concern by pasting the second-best baseline code here.
>
> Python
>
> ```
> class MLPModel(nn.Module):
>     def __init__(self, input_dim=30, negative_slope=0.01):
>         super(MLPModel, self).__init__()
>         self.activation = nn.LeakyReLU(negative_slope=negative_slope)
>
>         # hidden layers
>         self.hidden_layers = nn.Sequential(
>             nn.Linear(input_dim, 192),
>             self.activation,
>             nn.Linear(192, 160),
>             self.activation,
>             nn.Linear(160, 128),
>             self.activation,
>             nn.Linear(128, 160),
>             self.activation,
>             nn.Linear(160, 160),
>             self.activation,
>             nn.Linear(160, 32),
>             self.activation
>         )
>
>         # output layers
>         self.out1 = nn.Linear(32, 1)
>         self.out2 = nn.Linear(32, 1)
>         self.out3 = nn.Linear(32, 1)
>
>     def forward(self, x):
>         x = self.hidden_layers(x)
>         y1 = self.out1(x)
>         y2 = self.out2(x)
>         y3 = self.out3(x)
>         out = torch.cat((y1, y2, y3), dim=1)
>         return out
> ```
>
> It is so simple, yet its performance is so strong—stronger than the more sophisticated Transformer and AutoML baselines. This raises a severe doubt in my mind: How difficult is the task itself? Are the improvements achievable from simple engineering tricks, for example, slightly altering the structure by using `nn.LayerNorm`, changing activation functions, or adding residual connections to this MLP?
>
> Unfortunately, the authors refrain from replying some code, so I will not have an opportunity to verify this during the rebuttal process.
>
> **W3:** I agree that it is not necessary to embed strict physics constraints. However, the problem is the cross-attention module in equations 10-12 is so strong in its expressive power that it can learn any dependency and is not biased toward a more "physics correct" one. Mathematically, the gradients with respect to the model's core parameters are generally non-zero, meaning the model can always update its weights to fit any function. We all know "Attention is All You Need."
>
> But if you insist on this point, I'm open to it. I just think it is an overclaim. In this way, in geographic information science, tons of data can be splitted into multiple part, (e.g. population, economics, traffic dynamics), applying cross-attention, and we can claim it is guided by some physics or spatial-temporal dynamics, without really injecting the domain knowledge. You do similar things by using cross-attention and throwing the physics equations away. Is this reasonable or convincing?
>
> **Summary**
>
> To summarize, all of my concerns remain with your implementation, particularly:
>
> - The very little code in the repository.
> - The simple design of your model.
> - And the strong but very simple baselines.
>
> I confirm that there are no particulars in processing the 1-D sequential data, so I doubt that your task is not fundamentally different, your baselines are not extensively tuned, and the improvements can be achieved with engineering tricks to better fit the datasets. While your paper's story looks pretty good, I retain my doubts as aforementioned.

---

### Official Review · Reviewer_8XNb · 2025-07-02

**Clarity:** 3
**Significance:** 2
**Originality:** 3
**Rating:** 4
**Confidence:** 3

**Summary:**

This paper presents UCFormer, a physics guided deep learning model based on transformer architectures designed to capture urban climate processes and predict air temperature, dew point temperature, and specific humidity. The model includes an encoder that aims to capture how the urban surface modulates incoming shortwave and longwave radiation and surface fluxes within the urban canyon. This encoder returns a latent representation of the outcome variables: air temperature, dew point temperature, and specific humidity. A decoder then converts the latent representations of the outcome variables to their target values. The structure of these transformer-based encoders and decoders is designed to encourage the model to learn and use physical rules and relationships when modelling atmospheric forcing-urban surface interactions and  decoding latent representations of urban climate properties to target variables. The model is trained on simulated data generated by the Community Land Model Urban from six cities and evaluated on a test set of data from 2050-2055, a generalisability experiment with simulated data from 2070-2074, and a comparison with Flux tower measurements. Compared to a range of deep learning architectures, which were not designed to emulate urban climate processes, UCFormer generated more accurate predictions of the target variables on the test set and future generalisation set. The UCFormer was also fine-tuned to predict latent and sensible heat fluxes that are measured at Flux towers and compared to CMLU simulated fluxes. The UCFormer model was better correlated with Flux tower measurements than CMLU.

**Questions:**

There was also scope for more explanation of the results comparing model simulations to Flux tower measurements. As I understood what was presented, UCFormer was pre-trained using CMLU simulations and then calibrated (fine-tuned) with a subset of Flux tower measurements. Was the CMLU model also calibrated to Flux tower measurements? If not, are these results to be expected that UCFormer outperformed CMLU? Also, the model predictions of latent heat seem quite poor? Is there an explanation for this? And, how does this impact usability of the model or guide further research?

A suggestion is to provide more details of the training dataset’s characteristics. Specifically, what was the spatial resolution / detail of the input data and where they were sourced from (e.g. parameters and variables characterising the urban surface). Also, it was not clear what the spatial resolution of the output predictions was? Providing such information would be useful to gauge what use cases the presented model is relevant for.

I was interested in the distribution of predictions of target variables presented in the appendix. Have the author’s explored these in more detail and spatially and noted if there are certain geographic contexts / urban environments where model predictions are particularly erroneous? Relatedly, I would be interested in seeing results of the UCFormer predictions against observations from weather stations, and was there a reason the authors did not include such analysis as part of their evaluation?

**Ethical Concerns:**

["NO or VERY MINOR ethics concerns only"]

**Final Justification:**

I thank the authors for their discussion during the review. Overall, I have not changed my rating of the paper primarily due to the above concerns regarding the validation.

**Limitations:**

Yes (in the appendix)

**Quality:**

3

**Strengths And Weaknesses:**

The authors present a well thought through and described approach to integrate physical rules and understanding of urban climate systems into a machine learning framework. The description of the model was comprehensive and there was a thorough set of evaluations and ablations studies. However, there was scope for more discussion of how the model could be applied to guide management and planning of urban environments. For example, if the primary aim of the model presented here was to emulate a physical model simulations, more detail could have been provided on computational efficiency gains (beyond a single line in the introduction) and why such gains are necessary (e.g. what kinds of analyses that have relevance for mitigating urban heat could be undertaken with more computationally efficient models?). However, if the primary aim was to more accurately simulate key urban climate variables, was there a reason why the authors did not evaluate the model against ground truth measurements (e.g. from weather stations) and compare physical models to the presented ML model? The comparison of UCFormer against other, more general, ML models was interesting. However, the improved performance of UCFormer is to be expected given the architecture is tailored for this task. It would have been interesting to see how UCFormer compared against other urban climate models.

---

> ### Author Rebuttal · Authors · 2025-07-30
>
> **Q1: Explanation of the results comparing model simulations to Flux tower measurements.**
>
> **Response**:
>
> **1.Model performance on Flux tower measurements:** As correctly noted by the reviewer, UCformer was pre-trained on CLMU simulation data and then fine-tuned using a subset of Flux tower measurements. In this work, the CLMU model was not calibrated using these observations. It is important to note that CLMU is a state-of-the-art land surface model and ranks among the top performers in simulating Flux tower data across more than 30 models (Urban-PLUMBER project,  see reference [17] in the manuscript).
>
> Given that UCformer was fine-tuned with only a small amount of observational data, we are encouraged to see that it outperforms CLMU in this setting. This result is particularly noteworthy because calibrating physical models like CLMU to site-specific observations is computationally intensive. These results suggest that UCformer has strong potential to support real-world urban climate applications, especially in data-scarce environments where traditional model calibration is not feasible
>
> **2.Model Performance on Latent Heat:** Latent heat flux in urban areas is influenced by complex factors such as soil moisture, vegetation, and human activities like irrigation. Both physical and ML models struggle to simulate latent heat accurately, as also noted in the Urban-PLUMBER project (reference [17] in the manuscript), which benchmarks leading physical models. This difficulty arises because key variables such as vegetation and irrigation are often missing, uncertain, or represented in a highly simplified manner. In addition, unaccounted-for human influences further complicate the simulation.
>
> Future research could address this limitation by integrating the aforementioned variables into the model, provided that high-quality and complete observational data are available. In addition, explicitly guiding the model learning process with surface energy balance constraints may further improve ML model performance in estimating latent heat flux in urban environments.
>
> **Q2: Spatial resolution and characteristics of the training dataset.**
>
> **Response:**
>
> Our study focuses on learning the underlying physical processes governing urban climate, rather than making estimations at a fixed spatial resolution. This is because the training data were generated using the single-point mode of the physics-based CLMU model, which does not operate on a fixed resolution. Instead, its spatial footprint is determined by the resolution of the atmospheric forcing and surface input data. Specifically, during model development, we used forcing data from the atmospheric component of the Community Earth System Model at a native resolution of 0.9° × 1.25°. Urban surface characteristics were extracted from CLMU’s static surface input files at the same resolution. The CLMU output in single-point mode shares this spatial resolution, which aligns with the default resolution used in global urban climate projections.
>
> However, as shown in Section 4.6, we further evaluated both UCformer and the physical model using flux tower data with a much finer spatial footprint (~500 m),  demonstrating the model’s ability to generalize across scales. Overall, our framework is flexible and can incorporate higher-resolution forcing and surface data as they become available. With the growing availability of urban-scale reanalysis and detailed surface datasets, UCformer can be extended to simulate urban climate at finer spatial scales, supporting localized planning and adaptation analysis.
>
> **Q3: Data distribution and why not include weather station data for evaluation.**
>
> **Response:**
>
> **1.Data distribution and model spatial generalization:** Thank you for this insightful question. Our preliminary analysis shows that spatial generalization is highly challenging when training on a limited number of cities. Model performance depends on both urban surface features and regional climate. For instance, a model trained on London generalizes well to Edinburgh, likely due to their similar climates, despite differences in urban morphology. In contrast, performance drops in Shanghai, where both climate and surface characteristics differ markedly.
>
> Regarding whether model predictions are particularly erroneous in certain contexts, we recognize that standard metrics such as RMSE do not fully capture the model’s effectiveness in urban climate estimation. For example, in our original settings experiment, the temperature RMSE of Shanghai was 0.4437, while the MESS score was above 0.84. These high values indicate a substantial difference between urban and background temperatures in Shanghai. As a result, although RMSE reflects the overall prediction error, the MESS metric provides important context for assessing the model’s relative ability to distinguish urban climate from its background environment.
>
> Additionally, your concern is also of interest to other reviewers regarding the model’s performance on held-out cities. In response, we conducted additional "leave-one-out" experiments, as we intentionally chose these training cities with significantly different urban morphology or climate regimes. The latest results are provided below for your reference (original-setting results shown in parentheses).
>
> | 2050-2054     |                 | T               |                 | q               |                 | t               |                 |
> | ------------- | --------------- | --------------- | --------------- | --------------- | --------------- | --------------- | --------------- |
> | Held-out city | MESSagg         | RMSE(K)         | MESS            | RMSE(g/kg)      | MESS            | RMSE(K)         | MESS            |
> | London        | 1.6629 (1.9007) | 0.2990 (0.2419) | 0.8127 (0.8541) | 0.1344 (0.1117) | 0.4357 (0.5171) | 0.1977 (0.1655) | 0.4144 (0.5295) |
> | New York City | 1.6989 (1.8901) | 0.4167 (0.3515) | 0.7699 (0.8030) | 0.1904 (0.1645) | 0.5063 (0.5741) | 0.2507 (0.2212) | 0.4225 (0.5130) |
> | Shanghai      | 1.9369 (2.0024) | 0.4806 (0.4437) | 0.8349 (0.8486) | 0.2886 (0.2611) | 0.5555 (0.5986) | 0.2529 (0.2426) | 0.5465 (0.5552) |
> | Singapore     | 2.2840 (2.4586) | 0.2668 (0.2412) | 0.8581 (0.8727) | 0.2798 (0.2130) | 0.7060 (0.7975) | 0.1279 (0.1036) | 0.7198 (0.7884) |
> | Sao Paulo     | 1.8953 (2.1241) | 0.3965 (0.3561) | 0.8179 (0.8363) | 0.2587 (0.2074) | 0.5326 (0.6618) | 0.1649 (0.1435) | 0.5448 (0.6260) |
> | Rome          | 1.8163 (1.9048) | 0.3972 (0.3242) | 0.7843 (0.8282) | 0.1571 (0.1477) | 0.5197 (0.5450) | 0.1975 (0.1896) | 0.5123 (0.5316) |
>
> **2.Why not include weather station data for evaluation:** We did not include weather station data in our main evaluation for several reasons. First, weather stations are typically sited in non-urban environments (e.g., hilltops or airports) following World Meteorological Organization (WMO) guidelines, which aim to measure background weather conditions. As a result, their observations may not accurately reflect true urban climate characteristics. Second, while some low-cost sensors capture urban air temperature, it is often difficult to obtain detailed information on the surrounding urban surface characteristics. This limitation also motivated our use of physics-based simulations, which provide urban climate variables conditioned on specific atmospheric forcing and surface features.
>
> We fully acknowledge the importance of real-world evaluation. In this study, we used Flux tower observations as an alternative. Unlike typical weather stations, this Flux tower is fully embedded in an urban environment and provides both detailed atmospheric data and rich information on surrounding surface conditions. This aligns closely with the inputs to our model, making it a meaningful and appropriate benchmark. Therefore, we chose the Flux tower dataset rather than weather station data to evaluate real-world performance.
>
>
>
> **Weakness: The primary aim of the model and how UCformer could be applied to guide management and planning of urban environments.**
>
> **Response:**
>
> Our primary goal is to emulate a physical model simulation and learn urban climate processes. We agree that the manuscript would benefit from more discussion on computational efficiency and its practical relevance. Specifically, our model reduces the simulation time for a 55-year city-scale run from ~12 hours to ~17 seconds. This substantial speed-up enables applications previously constrained by computational cost. For example, with scenario-adaptive modules, UCformer could be fine-tuned to quickly assess urban climate responses to strategies such as green roofs, added vegetation, reflective surfaces, or altered building density. In optimization tasks requiring large-scale scenario testing, this efficiency offers clear advantages over conventional models, supporting planners and policymakers in adaptive urban management. We appreciate the reviewer’s suggestion and will include these points in the revised manuscript.
>
> Although our primary goal is to emulate physical model simulations, we also care about UCformer’s performance against real-world data. Therefore, in Section 4.6, we evaluated UCformer using ground truth measurements from Flux tower observations (please refer to our response to Q3 for the rationale behind not using weather station data). We chose CLMU as the comparison baseline in this part because it is a widely used and well-regarded physical model in urban climate research. In the Urban-PLUMBER project (reference [17] in the manuscript), which benchmarks leading physical models against Flux tower measurements, CLMU ranked among the top performers. Given its credibility and relevance, it serves as a robust baseline for evaluating UCformer’s performance against observational data.

---

> > ### Comment · Reviewer_8XNb · 2025-08-03
> >
> > I thank the reviewers for their very thorough response. Overall, I have not changed my original ratings of the paper. I still have questions regarding the motivation and intended application of this model. For example, as stated in the response, the model is trained to run for a single point with an approx. 100 km x 100 km footprint, it is not clear how the model could be used in scenarios suggested “green roofs, added vegetation, reflective surfaces, or altered building density”. This could be explained to highlight the utility of the model. I would be interested in the author’s thoughts on the relative needs for developing a relatively coarse scale urban climate model, as presented here, versus emulating microclimate models that relate to the scale or urban processes and planning / management interventions on-the-ground. My comments regarding validation of the model remain. As the model is trained to emulate an urban climate model, and validated against model outputs, it is not clear how well the model performs over actual cities. It is good that the authors evaluate the model using a Flux tower, but that is a single location and insufficient to understand the model’s general performance. I would encourage the authors to consider alternative ways of validating the model. I note the author’s comments that many weather stations are located over airports, but there are many cities around the world where there are weather stations with data availability that are located within in urban areas. There is also a body of work cleaning crowd-source weather station data for monitoring urban climates that could be alternative data source.

---

> > > ### Author Response · Authors · 2025-08-06
> > >
> > > **Response:**
> > >
> > > We thank Reviewer 8XNb for the continued engagement and thoughtful feedback. Noting the two major concerns that remain, we would like to provide the following clarifications:
> > >
> > > **1.How the relatively coarse-scale urban climate model could be used in urban climate adaptation and why it is important compared to microclimate models.**
> > >
> > > **(1) Model spatial resolution:** As noted in our previous response to Q2, both the CLMU single-point mode and the trained emulator can, in principle, operate at any spatial resolution, as long as the necessary input data are available. In this sense, spatial resolution can be viewed as part of the model configuration. We have demonstrated its application within the footprint of a flux tower (~500m radius), which showcases its potential adaptability to finer spatial scales.
> > >
> > > **(2) Why relative-coarse model important and how it could be used in urban climate adaptation:** Overall,  coarse-scale and microclimate models serve complementary roles in urban adaptation research. Simulations at the 1° (~100 km × 100 km) scale are a common and effective practice in the global urban climate community [1–3]. Such coarse-scale modeling can robustly evaluate the large-scale impacts and regional variability of climate adaptation measures across diverse cities and climates, directly supporting broad policy and planning decisions.
> > >
> > > For example, previous studies using CLMU at approximately 100 km resolution have demonstrated that implementing cool roofs was found to significantly reduce annual mean urban temperatures in regions such as the United States and China, whereas comparable benefits were not clearly observed in Europe or India [3]. Furthermore,  increasing roof albedo can significantly reduce the urban heat island effect in most regions, except for less urbanized areas in Africa and Mexico [3].  Specifically, some studies highlight that cool roofs tend to be most beneficial in areas characterized by higher solar radiation, lower precipitation, and weaker winds, while less urbanized or high-latitude regions may benefit less or differently [2-3].
> > >
> > > Such coarse-scale modeling enables researchers and policymakers to understand and compare these regional variations effectively, facilitating more informed, targeted, and context-sensitive urban adaptation strategies. This approach has become well established and is widely acknowledged within the urban climate science community.
> > >
> > > References:
> > >
> > > [1] Zhao L, Oleson K, Bou-Zeid E, et al. Global multi-model projections of local urban climates[J]. Nature Climate Change, 2021, 11(2): 152-157.
> > >
> > > [2] Wang L, Huang M, Li D. Where are white roofs more effective in cooling the surface?[J]. Geophysical Research Letters, 2020, 47(15): e2020GL087853.
> > >
> > > [3] Zhang J, Zhang K, Liu J, et al. Revisiting the climate impacts of cool roofs around the globe using an Earth system model[J]. Environmental Research Letters, 2016, 11(8): 084014.
> > >
> > > **2.Validation of the model against weather station data**
> > >
> > > We sincerely thank the reviewer for this valuable suggestion; validating our model with additional real-world data would certainly strengthen our results. We have carefully considered the idea of using weather station data for model validation, as recommended. However, we would like to further clarify the reasons why, in the current context, such validation using available weather station data is not feasible.
> > >
> > > To the best of the knowledge of the urban climate scientists among the authors, there are currently no harmonized, publicly accessible datasets of “urban weather stations”. Even in the rare cases where such stations exist in true urban environments, their sparse distribution, often just one or two stations per city, prevents them from representing the city-wide climate. Their measurements are dominated by the pronounced heterogeneity of the intra-urban landscape, and the absence of detailed accompanying urban surface parameters makes such data unsuitable for the validation required in this work. This persistent lack of urban observational data has been a major obstacle for urban climate research for decades, limiting observation-based studies and constraining the validation of both climate and weather models. Consequently, this data scarcity motivates us to use and publicly release physics-based simulation datasets to help compensate for the lack of real-world urban climate observational data.
> > >
> > > Additionally, while some crowd-sourced weather data exist for urban areas, these datasets are managed by various platforms and companies, each with different data quality, formats, and standards for processing and storage. Integrating such heterogeneous, multi-source crowd-sourced data into a unified, ML-ready dataset would be a highly interesting and valuable undertaking—one that we plan to explore in our future work.

---

### Official Review · Reviewer_PhTs · 2025-07-23

**Clarity:** 3
**Significance:** 3
**Originality:** 3
**Rating:** 4
**Confidence:** 1

**Summary:**

This paper introduces UCformer, a multi-task, physics-guided Transformer architecture tailored for modeling urban climate dynamics. The model jointly predicts 2-m air temperature, specific humidity, and dew point temperature, embedding domain-specific urban surface-atmosphere interactions and physical priors into its architecture. The authors argue that conventional ML approaches inadequately capture complex urban processes due to a lack of physical grounding. UCformer addresses this gap via a domain-specific encoder and a physics-guided decoder, aiming to enhance generalizability and interpretability. Extensive experiments on both simulated and real-world datasets show that UCformer outperforms baseline models in predictive accuracy and generalization capacity, and can adapt to data-scarce real-world environments with minimal fine-tuning.

**Questions:**

Comparison to physics-guided ML: As the authors suggested, there are several physics-guided ML models, in particular in climate modeling. How does UCformer compares to these previous models? Are there some obvious reasons that these previous models are not applicable to urban climate modeling?

Real-World Evaluation: How well does UCformer generalize to cities not included in training (e.g., cities with significantly different urban morphology or climate regimes)?

Model Efficiency: Can the authors comment on inference time or model size relative to other ML-based climate models?

**Ethical Concerns:**

["NO or VERY MINOR ethics concerns only"]

**Final Justification:**

The rebuttal resolves the core issue for the above weakness, so I raised my score.

**Limitations:**

Yes

**Quality:**

3

**Strengths And Weaknesses:**

Strengths:

In terms of quality, the architecture is well-motivated, addressing an underexplored niche (urban climate modeling). Also, the experimental design is solid with diverse evaluation: emulation, generalization (future years), and transfer to real-world observations. There are clear performance gains over relevant baselines (Transformer, AutoML, MLP). Ablation studies reinforce the contribution of each module (encoder/decoder).

In terms of clarity, the paper is overall well-written.

In terms of significance, the paper addresses a critically understudied problem—urban climate prediction—with clear real-world implications (urban heat islands, planning). The paper could be highly impactful in climate-aware urban planning, if extended and validated further.

In terms of originality, embedding physics-informed priors directly into Transformer architectures in urban climate modeling is novel. The paper introduces a shared latent representation approach that exploits cross-city correlations, which is rare in ML climate literature.


Weaknesses:

I think the biggest weakness is that the paper lacks of detailed comparison to other physics-informed ML models, particularly in domains with overlapping temporal and spatial modeling needs. In Related work the authors have suggested several work on physics-guided ML, and hence the authors should have compared UCformer with previous work, or expain explain in the related work why previous work is not adequate for urban climate modeling.

---

> ### Author Rebuttal · Authors · 2025-07-30
>
> **Q1: Lack of comparison to other physics-informed ML models**.
>
> **Response:**
>
> We thank the reviewer for raising this important point. As discussed in the Related Work section, prior physics-guided ML models often incorporate explicit physical equations into the loss function or model architecture to enhance generalization and interpretability. For example, Read et al. (reference [27] in the paper ) employ a simplified energy budget formulation in which each flux term can be derived either directly from the ML inputs or from a combination of inputs and ML-predicted surface water temperatures. When the predicted terms fail to satisfy energy balance, the model is penalized accordingly during training. Similarly, in the physics-informed model proposed by Zanetta et al. (reference [41] in the paper ), a hard physical constraints layer is introduced. This layer takes a subset of predicted target variables, such as temperature and air pressure, to derive other physically related quantities, such as humidity. The model then computes a composite loss over all variables, encouraging physically consistent predictions. In other words, these physics-informed models are typically highly data- and task-specific. However, as noted in our manuscript (line 182), our study focuses on estimating three key variables in urban climate—2-m air temperature, dew point temperature, and specific humidity. Due to the lack of critical variables such as urban canyon air pressure, it is not feasible to directly embed relevant physical equations into the loss function or network structure for our tasks.
>
> As a result, it is not feasible to apply such physics-informed constraints in our setting, making these previous models unsuitable as baselines. Notably, our “softy” design, which does not depend on explicit physical constraints, enables broader applicability to diverse datasets, as shown in Section 4.6.
>
>
>
> **Q2: How UCformer generalize to cities not included in training.**
>
> **Response:**
>
> Spatial generalization remains quite a challenging issue in global climate and urban climate prediction/estimation. While our work primarily focuses on improving temporal generalization, we agree that evaluating spatial generalization is crucial for real-world applications. To provide further insight, we conducted additional “leave-one-out” experiments. When selecting training cities, we intentionally chose cities with significantly different urban morphology or climate regimes. Therefore, to better assess spatial generalization, we performed supplementary experiments where one city was held out during training. The results are provided below for your reference (results in parentheses indicate performance under the original setting).
>
> | 2050-2054     |                 | T               |                 | q               |                 | t               |                 |
> | ------------- | --------------- | --------------- | --------------- | --------------- | --------------- | --------------- | --------------- |
> | Held-out city | MESSagg         | RMSE(K)         | MESS            | RMSE(g/kg)      | MESS            | RMSE(K)         | MESS            |
> | London        | 1.6629 (1.9007) | 0.2990 (0.2419) | 0.8127 (0.8541) | 0.1344 (0.1117) | 0.4357 (0.5171) | 0.1977 (0.1655) | 0.4144 (0.5295) |
> | New York City | 1.6989 (1.8901) | 0.4167 (0.3515) | 0.7699 (0.8030) | 0.1904 (0.1645) | 0.5063 (0.5741) | 0.2507 (0.2212) | 0.4225 (0.5130) |
> | Shanghai      | 1.9369 (2.0024) | 0.4806 (0.4437) | 0.8349 (0.8486) | 0.2886 (0.2611) | 0.5555 (0.5986) | 0.2529 (0.2426) | 0.5465 (0.5552) |
> | Singapore     | 2.2840 (2.4586) | 0.2668 (0.2412) | 0.8581 (0.8727) | 0.2798 (0.2130) | 0.7060 (0.7975) | 0.1279 (0.1036) | 0.7198 (0.7884) |
> | Sao Paulo     | 1.8953 (2.1241) | 0.3965 (0.3561) | 0.8179 (0.8363) | 0.2587 (0.2074) | 0.5326 (0.6618) | 0.1649 (0.1435) | 0.5448 (0.6260) |
> | Rome          | 1.8163 (1.9048) | 0.3972 (0.3242) | 0.7843 (0.8282) | 0.1571 (0.1477) | 0.5197 (0.5450) | 0.1975 (0.1896) | 0.5123 (0.5316) |
>
> The results show that UCformer exhibits a certain degree of spatial generalization, but its performance varies notably across cities and is strongly influenced by both urban morphology and local climate conditions. For instance, the RMSE of temperature estimates ranges from approximately 0.02 K in Singapore to 0.07 K in Rome, while the RMSE of dew point temperature estimates ranges from about 0.01 K in Rome to 0.03 K in London.
>
> Overall, these findings suggest that while UCformer is capable of spatial generalization, its performance is influenced by the uniqueness of the target city’s urban and climatic conditions. We acknowledge spatial generalization as an important direction for future research. In particular, we plan to further investigate the shared representations of urban climate processes proposed in Section 4.5, and aim to explore whether scaling laws emerge with respect to the number and diversity of cities included during training, and how these factors affect the model's spatial generalization capability across heterogeneous urban environments and climate regimes.
>
>
>
> **Q3: Model Efficiency.**
>
> **Response:**
>
> In addition to UCformer, we evaluated two representative sequence-to-sequence models, that is, LSTNet and Informer, which can be adapted to urban climate estimation.
>
> All models were tested with a batch size of 128 on an NVIDIA 5090 GPU. The inference time for UCformer on our test set (219,000 data points) is approximately 1.4 seconds, with a model size of about 9.35 million parameters. In comparison, Informer requires about 3.1 seconds for inference and has 11.38 million parameters, while LSTNet is more lightweight with 111.13 thousand parameters and an inference time of about 2.3 seconds. The inference efficiency (covering 5-year simulations for 6 cities) of the current model meets the requirements for operational urban climate applications.
>
> These results indicate that, compared to Informer (a transformer-based model), our model has fewer parameters and a shorter inference time. Relative to a smaller LSTNet, which utilizes a CNN+RNN architecture, UCformer also demonstrates an advantage in inference efficiency.

---

> > ### Comment · Reviewer_PhTs · 2025-08-09
> >
> > I would like to thank reviewer for a thoughtful rebuttal.
> >
> > Q1: My main motivation for my score was the nonexistence of comparisons to physics-guided ML. And now I understand why prior physics-guided ML models are unsuitable for this setting, given the lack of necessary variables to embed explicit physical equations. I would suggest adding this reasoning more explicitly in the manuscript.
> >
> > Q2: The new leave-one-out results show UCformer can generalize spatially, though performance varies by city, influenced by morphology and climate. I would suggest briefly includingthese findings in the manuscript.
> >
> > Q3: The efficiency comparison shows UCformer is both faster and smaller than Informer and competitive with lighter models like LSTNet, while retaining accuracy. Including these metrics in the paper would highlight its practical advantages.
> >
> > Overall, the rebuttal resolves the core issues, so I would raise my score.

---

> ### Author Response · Authors · 2025-08-06
>
> Dear Reviewer PhTs,
>
> Thank you once again for your thorough review and insightful suggestions. Your feedback has been invaluable in guiding the refinement and improvement of our work.
>
> In response to your comments, we have provided a detailed explanation of why previous physics-informed ML models are not suitable for inclusion as baselines in this work. Additionally, we have conducted a significant new experiment using the “leave-one-out” strategy to evaluate the generalization capability of UCformer, particularly in cities with markedly different urban morphology or climate regimes. We have also added a discussion comparing the efficiency of our model with other ML-based climate models.
>
> We are actively participating in the Author-Reviewer Discussion phase and are happy to provide any further clarification as needed.
>
> Best regards,
>
> The Authors of Submission 2584

---

### Note · Authors · 2025-08-13

Dear Reviewers, AC/SAC/PC,

We sincerely thank all reviewers for their constructive feedback and valuable discussions, which helped us improve the paper. Our rebuttals addressed the concerns of `PhTs` and `whR3` and provided further comments on `8XNb` 's remaining concerns, with no additional feedback received. This final remark responds to `CyZa` ’s last concerns:

- The simplicity of the code repository.

- The simplicity of the model design.

- The strength of the baselines.

We respectfully disagree with these points and respond below:

**Code**: `CyZa` questioned the task’s difficulty based on the simplicity of the baseline code. We consider this judgment unfounded, as code length or complexity is not a valid measure of a method’s validity or task difficulty. Model performance should be evaluated based on results and replicability, not code size.

We have released the necessary code, data, and checkpoints, with complete implementation details. The reviewer has not provided any concrete evidence to challenge the validity of our implementation or results.

**Model**: `CyZa` criticized the simplicity of our model while ignoring its strong performance. We believe the emphasis on model complexity is unwarranted, as simplicity is often a strength, particularly in physics-guided AI. The claim that cross-attention invalidates the physics-guided nature is also inaccurate. We carefully integrate domain and physical inductive biases into neural networks, guiding learning through interactions with physically meaningful latent representations.

Besides, `CyZa` repeatedly framed our task as standard *time series forecasting (TSF)*. However, our objective is fundamentally different—it aims to understand *feature–target relationships*, not to predict future values. Although Transformers are used in our model, this does not equate the task with *TSF*.

**Baseline**: `CyZa` questioned the strength of our baselines. However, the baseline’s strength is consistent with results from major climate benchmarks like *ClimSim*. We also added experiments using modified SOTA *TSF* models as requested, which underperformed our strong baselines. The reviewer’s insistence on such methods reflects biases regarding our task.

We appreciate all reviewers' engagement and feedback. We are especially grateful to those who recognized the merits of our interdisciplinary AI-for-science contribution.

Thank you for your time and consideration.

Best,

The Authors of Submission 2584

---

### Decision · Program_Chairs · 2025-09-17

**Decision:**

Accept (poster)

**Comment:**

This paper introduces UCformer, a physics-guided transformer model for urban climate emulation, jointly predicting air temperature, dew point, and specific humidity. The authors release a valuable new simulation-based dataset spanning multiple cities and decades. Experiments demonstrate improved performance compared to some baseline ML methods and show UCformer can be fine-tuned to real-world data (from urban Flux tower) with competitive results against leading physical models.

The paper tackles an important, underexplored problem and presents a solid AI approach with good theoretical motivations. The released dataset is likely to be highly useful for future research. Reviewers noted the interdisciplinary value of the work.

Likely due to the modern transformer focused context, the authors failed to recognize reviewer CyZa's key point about under-engagement with the rich literature on time-series "forecasting". Might have been useful to instead frame it as time-series representations/feature-engineering and lack of such comparison can make it unclear whether this will have real world impact. Many of the "transformer" time-series models have the same problem as well, there is very little underlying data to really not just be overfitting with transformer style methods (which are very good at it without proper regularization under limited data constraints) which often do well on benchmark time-series tasks, but not in the real world. Given the main results are on simulated data with only limited observation-based evaluation (as highlighted by several reviewers), this raises even more concerns about generalization and impact as it's much easier to have train/test distribution easily match up, compared to the real world. I strongly encourage the authors, in the final version of this paper, to engage more deeply with the substantive criticisms raised by reviewer CyZa.

The rebuttal phase was constructive. Authors provided detailed responses on spatial generalization, model efficiency, and tried to clarify the rationale behind their choices. I appreciate the authors' commitment to code and data release making it easier for the community to run more experiments in this domain.